# Feeding Value of Lupins, Field Peas, Faba Beans and Chickpeas for Poultry: An Overview

**DOI:** 10.3390/ani14040619

**Published:** 2024-02-14

**Authors:** Laura S. David, Catootjie L. Nalle, M. Reza Abdollahi, Velmurugu Ravindran

**Affiliations:** 1Monogastric Research Centre, School of Agriculture and Environment, Massey University, Palmerston North 4442, New Zealand; l.david@massey.ac.nz (L.S.D.); catootjienalle@gmail.com (C.L.N.); m.abdollahi@massey.ac.nz (M.R.A.); 2Animal Husbandry Department, Polytechnic of Agriculture Kupang, Prof. Herman Yohannes St., Lasiana, Kupang 85228, NTT, Indonesia; 3A2Z Poultry Feed DynamikZ, 69100 Villeurbanne, France

**Keywords:** antinutrients, feeding value, feed processing, grain legumes

## Abstract

**Simple Summary:**

The fluctuating supply and price of soybean meal have steered nutritionists to explore alternative protein sources in poultry diets. An attractive option is grain legumes because of their reasonable protein, amino acid and energy contents. The presence of antinutritional factors in grain legumes is a major concern associated with their use in poultry feeding. However, as documented, there are effective processing methods that can mitigate the negative effects of these antinutritional factors. The current overview deals with the feeding value of four selected grain legumes, namely lupins, field peas, faba beans and chickpeas, in poultry feeding.

**Abstract:**

Grain legumes are fair sources of protein, amino acids and energy, and can be used as a replacement for soybean meal in poultry feed formulations as the soybean meal becomes short in supply and costly. However, a concern associated with the use of grain legumes in poultry feeding is the presence of antinutritional factors. The effective processing and utilisation of these grain legumes in poultry feeding are well documented. The current review focuses on four selected grain legumes (lupins [*Lupinus albus* and *Lupinus angustifolius*], field peas [*Phaseolus vulgaris*], faba beans [*Vicia faba*] and chickpeas [*Cicer arietinum*]) and their nutrient content, the presence of antinutritional factors, processing methods and feeding value, including updated data based on recent research findings.

## 1. Introduction

The global poultry industry relies mostly on soybean meal (SBM) and meat and bone meal as protein sources in diet formulations. However, the fluctuating supply and high price of SBM and the ban on the use of meat and bone meal in some regions of the world have become major challenges to the industry. Efforts to address these challenges have provided impetus for the greater exploitation of alternative protein sources which can fully or partially replace these conventional protein sources. There are number of acknowledged limitations of alternative ingredients, which preclude their use by the commercial industry; these include nutritional, technical and socio-economic issues as outlined by Ravindran and Blair [1]. Among the potential alternatives, grain legumes are promising to be used as substitutes for conventional protein feed sources.

Grain legumes, often referred as pluses, are the non-oilseed, dry, nutritionally dense, edible seeds of leguminous crops within the family Fabaceae or Leguminosae. Unlike other legumes, grain legumes remain on plants to dry before harvesting [2]. According to Sipas et al. [3], the use of the term ‘pulse’ is associated with human nutrition, whereas the term ‘grain legume’ is generally allied with the feed industry. The Food and Agriculture Organisation (1994) recognises 11 types of grain legumes (Table 1): dry beans, dry broad beans, dry peas, chickpeas, cow peas, pigeon peas, lentils, Bambara beans, vetches, lupins, and pulses nes (‘not specified elsewhere’ or minor pulses that do not fall into one of the other categories).

The current overview is not intended to provide an analysis of the feeding value of all 11 major grain legumes for poultry. The focus will be on four species, namely lupins, field peas, faba beans and chickpeas, which are considered to have the greatest potential in terms of agronomy, seed yields and animal nutrition in the Australasian region [5,6,7,8]. These grains, in general, are moderate to good sources of protein containing 200–400 g/kg crude protein. Their essential amino acid (AA) profile is high in lysine and low in methionine. There is a large volume of published data on the nutrient composition of these four species, but reports on the bioavailability of energy and AAs remain limited and scattered. The current overview is an attempt to collate these data into one publication. First, we present a compilation of variability that exists in the nutritional value of each grain legume. During the past two decades, ileal-based AA digestibility is being increasingly accepted and has become the norm in poultry feed formulations [9,10]. Recently published data on the ileal AA digestibility, along with apparent metabolisable energy (AME), and results of feeding trials, are reviewed. Second, we provide a brief summary of antinutritional components that interfere with nutrient availability and whose presence is responsible for limiting the inclusion rates of grain legume in practical diets. Last, we discuss the various strategies to remove or lower the contents of antinutritional factors to mitigate the adverse effects on poultry.

## 2. Nutritional Value

### 2.1. Lupins (Lupinus spp.)

*Lupinus* is a large genus that has more than 300 species in both the Eastern and Western Hemispheres. Lupins are native to North and South America, the Mediterranean region and northern Africa. Only five species [11], however, are cultivated and they are *L. albus* (white lupin)*, L. angustifolius* (narrow-leaf lupin)*, L. luteus* (yellow lupin)*, L. mutabilis* and *L. cosentinii* (sandplain lupin). Of these five species, the first three are suitable for cultivation as high-protein crops [12]. Based on their promise in Australasia, only the first two species are reviewed herein.

Lupin production was initially limited to white lupin cultivars. The interest in using narrow-leaf lupins as an alternative to conventional protein sources in poultry diets has been increasing in recent decades, especially in Australia. Currently, Australia is the largest lupin grain producer in the world, and the narrow-leaf lupin is the dominant species. Lupin seeds are an attractive alternative to soybeans because of their high protein content (202–424 g/kg; Table 2).

Older lupin cultivars contain various types of alkaloids of which the quinolizidine alkaloids are the most relevant antinutritional factor. Alkaloids are defined as nitrogen-containing water-soluble compounds produced in the chloroplasts of some plants with the purpose of repelling insects [13]. Lupanine is the major alkaloid present in *L. albus* and *L. angustifolius,* while lupinine is present in *L. luteus* [14]. Some other alkaloids such as sparteine, angustifolin and gramine are also present in *L. luteus*. Based on the alkaloid content, lupin can be grouped into two categories: those with a high alkaloid content (up to 53.8 g/kg), commonly known as bitter lupins, and those with low alkaloid content (less than 0.5 g/kg), referred to as sweet lupins [13]. Sweet lupins can either be of the white (*L. albus*), yellow (*L. luteus*) or blue-seeded (*L. angustifolius*) cultivars [14]. Early cultivars of lupins contained relatively high concentrations of toxic and bitter alkaloids that depressed feed intake and growth, and negatively affected the feed efficiency in broilers [15,16]. However, modern plant breeding techniques have now enabled the development of low-alkaloid lupin cultivars. For example, current Australian sweet lupins are known for their virtually zero alkaloid content (less than 0.4 g/kg; [3]).

#### 2.1.1. *Lupinus angustifolius*

This lupin species is referred to as narrow leaf lupin, narrow-leaved lupin or blue lupin. This is an annual herb that can reach 80 cm or more in height. The inflorescence of sweet lupins bears many flowers that are usually blue in colour but can also range from white to pink [12]. The seeds of sweet lupin have variable colours from dark gray to brown to white, and can be speckled or mottled. As noted above, cultivars with a low alkaloid content are called sweet lupins. This species contains a single recessive gene that controls sweetness. The bitter form of the gene causes seeds to have a high alkaloid content that can be poisonous and cause liver damage. For almost a century, plant breeders have been developing cultivars with lower alkaloid content. Culvenor and Petterson [17] reported that some sweet lupins can contain as low as 0.02 g/kg alkaloids. The alkaloid content of bitter lupins could be 1000 times greater than sweet lupins, but these cultivars have a higher seed yield [12].

In Australia, sweet lupins dominate the commercial market. The nutritional composition of sweet lupins is acknowledged by the feed manufacturers. However, the nutritional variability between cultivars [18,19,20] is a major challenge. Reported analysis for the crude protein content of sweet lupins ranges from 223 to 409 g/kg dry matter [DM] (Table 2). This variation is largely caused by differences in cultivars, production location and year, and agronomic management [20,21].

**Table 2 animals-14-00619-t002:** The nutritional composition (g/kg, dry matter basis) of Australian sweet lupins.

Nutrient	Average	Range *	References
Dry matter	916	889–957	[3,13,20,22,23,24,25,26,27,28,29,30,31,32]
Crude protein	328	223–409	[3,13,20,22,23,24,25,26,27,28,30,31,32,33,34,35,36]
Crude fat	68	43–81	[3,13,20,22,23,25,26,27,28,30,32,33,34,35,36]
Crude fibre	179	140–213	[3,13,24,27,28,30,33,36]
Acid detergent fibre	224	198–258	[3,22,24,25,26,27,36]
Neutral detergent fibre	272	240–307	[22,24,25,26,27,36]
Soluble fibre	34	34	[34]
Insoluble fibre	488	488	[34]
Ash	34	21–45	[3,22,23,24,25,26,27,28,30,33,34,36]
Starch	6.6		[27]
Calcium	2.2	1.9–2.4	[3,13,27]
Phosphorus	4.0	3.3–5.0	[3,13,23,27]

* Range is based on the average values reported in the given references.

Kingwell [21] reported that the protein and oil contents of sweet lupins are related to seed size. There was a tendency for bigger seeds to have higher protein and oil contents compared to the smaller seeds in the same cultivar. In comparison with field peas and faba beans, which contain more than 300 g/kg starch, the starch content of lupins is very much lower. Some lupin cultivars are reported to be completely devoid of starch [37]. The carbohydrate profile of lupins is dominated by structural carbohydrates, neutral detergent fibre and acid detergent fibre. The high soluble oligosaccharide content restricts wider acceptance of sweet lupins in poultry diets [38].

The proportion between hull and kernel, and their nutrient composition differ depending on the species of lupins [21]. The proportion of the seed coat in sweet lupins is about 230 g/kg. The seed coat contains mainly cellulosic fibre, while kernels comprise 300 g/kg cell wall materials, and pectin-like dietary fibres.

Published data on the AA content of Australian sweet lupins is summarised in Table 3. The AA profile is similar to other legume proteins, being high in lysine and low in sulphur-containing AAs and tryptophan. These limiting AAs can be supplemented with synthetic forms in diets containing sweet lupins. There may also be possibilities to increase the AA content through molecular techniques. It is worth noting that high-methionine transgenic lupins, containing 4.5 g/kg methionine, have been developed in Australia (Table 3).

##### Apparent Metabolisable Energy

The AME of sweet lupins differs between cultivars (Table 4), from 6.04 to 11.64 MJ/kg DM basis. Hughes et al. [39] reported that the AME of a cultivar (cv. Gungurru) of Australian sweet lupins from three Western Australian sites ranged from 9.8 to 12.3 MJ/kg. Observed variation within a cultivar reflects the differences in climate, soil and agronomic conditions. The low energy utilisation may be explained by the high content of non-starch polysaccharides (NSPs; soluble and insoluble) and extremely low content or lack of starch.

##### Amino Acid Digestibility

Available data on the apparent ileal AA digestibility coefficient of sweet lupins for broilers are summarised in Table 5. The AA digestibility in Australian sweet lupins is high and similar to those reported for SBM [10,26]. A lysine digestibility of 0.87–0.91 was reported in caecectomised laying hens [46].

##### Feeding Trials

Early research indicated that sweet lupins are not a suitable protein source in broiler diets. Olkowski et al. [16] showed the negative effects of feeding 350–400 g/kg sweet lupins (raw or dehulled or autoclaved; cv. Troll) on growth performance in young broilers and suggested that the substitution of lupin seed meal for SBM in broiler diets is only possible for broilers aged 4 weeks and above. It was speculated that the levels of alkaloids may have been responsible. Similarly, other early studies [47,48] have shown that the use of 200 g/kg sweet lupins reduced the growth and feed efficiency of broiler starters.

**Table 5 animals-14-00619-t005:** Apparent ileal amino acid digestibility coefficient of Australian sweet lupins.

Amino Acids	References						
	[26] ^1^	[27] ^2^	[32] ^3^	[43] ^4^	[44] ^5^	[49]	[50]
Essential							
Arginine	0.90	na	0.94	0.84	0.93	0.89	0.93
Histidine	0.84	0.84	0.79	0.76	0.86	0.84	0.78
Isoleucine	0.82	0.84	0.85	0.83	0.87	0.81	0.82
Leucine	0.84	0.81	0.87	0.84	0.88	0.83	0.85
Lysine	0.78	0.85	0.87	0.82	0.89	0.83	0.84
Methionine	0.83	0.85	0.79	na	0.79	0.82	0.76
Phenylalanine	0.83	0.82	0.89	0.83	0.85	0.83	0.87
Threonine	0.76	0.78	0.82	0.76	0.80	0.77	0.79
Tryptophan	0.79	0.76	na	na	na	na	na
Valine	0.80	0.77	0.83	0.80	0.89	0.80	0.80
Non-essential							
Alanine	0.80	na	0.83	0.81	0.87	0.80	0.82
Aspartic acid	0.82	na	0.84	0.78	0.87	0.82	0.81
Cysteine	0.69	na	0.83	na	0.77	0.78	0.82
Glycine	0.82	0.80	0.82	0.75	0.83	0.82	0.81
Glutamic acid	0.89	na	0.91	0.87	0.92	0.86	0.90
Proline	na	na	0.82	0.80	0.82	na	0.80
Serine	0.81	na	0.82	0.77	0.82	0.80	0.79
Tyrosine	0.85	0.79	0.84	0.78	0.76	0.83	0.84

^1^ cv. Warrah; ^2^ cv. Mandelup; ^3^ Average of three cvs. (Wallan, Tanjil and Borre); ^4^ Average of four cvs. (Sonet, Boruta, Graf and Neptun); ^5^ cv. Pershatsvet; na = not available.

A number of other studies, on the other hand, have demonstrated that sweet lupins can be safely used in poultry diets. The observed discrepancy may be explained by cultivar differences in the contents of alkaloids and NSPs, and the failure to consider the low AME in feed formulations. Nalle et al. [32] reported that narrow leaf lupins (cv. Wallan, Tanjil and Borre) can be included at 200 g/kg in broiler starter diets when the diets are properly balanced for AME and digestible AAs. Farrell et al. [47] studied different inclusion rates of lupins and suggested an inclusion level of less than 100 g/kg for broilers. van Barneveld [38], in contrast, indicated that lupins could be used in broiler diets up to 250 g/kg. Perez-Maldonado et al. [25] did not find any negative effect from feeding sweet lupins at 250 g/kg on the performance of laying hens when compared to field peas and faba beans. However, the same study reported an increased digesta viscosity and weight of pancreas at 250 g/kg sweet lupin. Perez-Escamilla et al. [51] similarly found that lupin inclusion level of 300 g/kg could support the performance of broilers without any detrimental effects. According to Hughes et al. [40], whole seeds of lupins can be included up to 200 and 300 g/kg in wheat-based and maize-based diets, respectively, for broilers. Brand et al. [52] reported that SBM can be replaced with sweet lupins up to 300 g/kg in the diet of grower ostriches. It is, however, worth noting that higher inclusion levels of lupins could increase the incidence of wet litter [38,40]. At 200 g/kg inclusion, the excreta quality was not affected [32].

#### 2.1.2. *Lupinus albus*

This lupin species is commonly known as white lupin or field lupin. The colour of the white lupin flowers are greyish-blue or white. This species is mainly distributed around the Mediterranean region, Europe, South America and tropical and southern Africa [53,54]. Seeds of white lupin are large, flat, rectangular or square-shaped with rounded corners, compress laterally and are about 7–16 mm long and 6–12 mm high [55].

The alkaloid content of bitter cultivars ranges from 5 to 40 g/kg, while those of low-alkaloid cultivars range between 0.08 and 0.12 g/kg [54]. Alkaloid-free cultivars of white lupins are also available, and the development of these alkaloid-free mutants has allowed the exploitation of white lupins as a protein source for animals.

White lupins contain moderate to high contents of crude protein (202–424 g/kg), crude fat (60–130 g/kg), and fibre content (105–162 g/kg) as summarised in Table 6. The considerable variation observed in the nutritional content of white lupins probably reflects genetic and environmental differences [53,54,55]. Brenes et al. [56] reported that the high portion of hull (16% of the seed) was mainly responsible for the high fibre content of the whole seed. Thus, the removal of the hull will markedly decrease the fibre content. White lupins have only a negligible amount of starch [57], but high amount of soluble and insoluble NSPs and oligosaccharides [27,38,42]. The oligosaccharide content is the feature which most often appears to limit their wider use in poultry diets.

Table 7 summarises the published data on the AA composition and indicates that white lupins are deficient in methionine, cysteine and tryptophan, but good sources of other essential AAs. Similar to sweet lupins [26], there may be possibilities to increase the content of methionine in white lupins through modern plant breeding techniques. However, to the best of authors’ knowledge, there are no published data on breeding techniques to improve the cysteine or tryptophan content of grain legumes. The AA composition of white lupins has been shown to differ from other lupin species (*L. angustifolius* and *L. luteus*) with higher concentrations of threonine, tyrosine and isoleucine [69]. In general, white lupin has higher AA (total and essential) content than Australian sweet lupins [24,27,30].

##### Apparent Metabolisable Energy

The AME values of white lupins have been reported to range from 8.1 to 13.3 MJ/kg (Table 8). The higher AME content of white lupins compared to Australian sweet lupins (Table 4) is due to their higher oil content [73].

##### Amino Acid Digestibility

Amino acids in white lupins are well digested (Table 9), with most AAs having digestibility coefficients of over 0.80.

##### Feeding Trials

The feeding value of lupins is determined, to a large extent, by the concentration of alkaloids in the seed. As discussed above, these bitter substances can influence the feed intake and growth in poultry and limit the utilisation of white lupins. However, with the development of new cultivars with low alkaloid content (<0.1 g/kg), this is no longer an issue.

Nalle et al. [75] reported that when balanced for AME and digestible AA, white lupins can be used at 200 g/kg in wheat–SBM and wheat–SBM–meat meal-based diets for broilers up to 35 days of age. Dietary lupin concentrations of 50–300 g/kg have also been shown to support the growth performance of broilers without any adverse effects [51,76]. Olver [77] reported that feeding broilers up to 8 weeks with 400 g/kg white lupins (alkaloid content, <0.1 g/kg) showed no adverse effects on growth, feed efficiency or carcass characteristics. Similarly, Olver and Jonker [71] reported that broiler chickens can tolerate up to 400 g/kg of white lupins (cultivar Hanti) without compromising their growth. A similar trend was also found in the feeding of ducklings [59] up to 6 weeks of age with diets containing up to 400 g/kg white lupins (cv. Buttercup). Increased egg yolk colour was reported in laying hens fed diets containing 100–300 g/kg lupins (cv. Ultra) [78]. It was concluded that this lupin cultivar could replace all the SBM in broiler diets and that white lupins do not exert any antinutritive effect provided that the concentration of alkaloids is less than 0.1 g/kg. In contrast, Olkowski et al. [79] reported a significant decrease in feed intake and weight gain in broilers fed a diet containing 400 g/kg raw white lupin seeds. This could be because of high-alkaloid content of white lupin. Feed intake and body weight are reduced with increasing dietary lupin concentrations (0.12–3.64 g/kg) as reported by Pastuszewska et al. [80]. The bitterness due to the high alkaloid content in lupins may reduce the feed intake and consequently the weight gain in birds. According to Kaczmarek et al. [81] and Kubiś et al. [82], the AME of diets linearly decreased with increasing inclusions of white lupins from 0 to 300 g/kg in the diets. Kaczmarek et al. [81] reported a growth depression in broilers fed the diets with >150 g/kg white lupins. A similar negative effect was also reported in turkey poults where there were 6, 11 and 15% reductions in the growth observed in 3-week-old poults fed diets with 300, 450 and 600 g/kg white lupin, respectively [70].

### 2.2. Field Peas

Field pea seeds can be smooth or wrinkled, and green, white or brown in colour. The average weight of seed is about 200 mg, with the seed coat contributing around 12% of the total seed weight [3]. The distinction between different field peas is made by the colour of the tegument (translucent without tannins and coloured with tannins) and the colour of the cotyledons.

Wide variability can be seen in the proximate composition of field peas (Table 10) and reflects the differences in cultivar, growing condition, and analytical methods. Field peas are a moderately high-quality source of protein and starch. Compared to SBM, field peas have lower protein content, ranging from 114 to 301 g/kg DM (Table 10). Field pea protein is reported to be highly digestible with an excellent AA balance [12]. Similar to other legumes, field peas are deficient in sulphur-containing AAs (Table 11). Lysine concentration is relatively high in field peas. The predominant fraction of field pea carbohydrates is starch, having an average content of 413 g/kg DM (Table 10). The fat content of field pea is very low (6.5–27 g/kg DM). The crude fibre content of field peas is higher (average of 101 g/kg DM) than that of SBM (38 g/kg; [82,83].

**Table 10 animals-14-00619-t010:** Nutritional composition (g/kg, dry matter basis) of field peas.

Nutrient	Mean	Range *	References
Dry matter	888	869–913	[3,22,24,25,31,50,58,84,85,86,87,88,89,90]
Crude protein	236	114–301	[3,22,24,25,31,33,50,58,84,85,86,87,88,89,90,91,92,93,94,95,96]
Crude fat	18	6.5–27	[3,22,25,33,50,58,85,86,87,88,89,90,93,94,96]
Crude fibre	101	49–286	[3,24,33,58,85,88,95,96]
Acid detergent fibre	85	100–230	[3,22,24,25,58,86,87,93]
Neutral detergent fibre	153	84–230	[3,22,24,25,50,58,86,87,93]
Starch	413	119–488	[58,87,89,90,92,93,94,95]
Ash	31	25–37	[3,22,24,25,33,50,58,85,87,89,90,93,94,96]
Calcium	0.9	0.5–1.2	[3,86,87,88,93]
Phosphorus	4.7	4.4–4.9	[3,86,87,88,93]

* Range is based on the average values reported in the given references.

#### 2.2.1. Apparent Metabolisable Energy

The reported AME values of field peas range between 8.3 and 12.3 MJ/kg (Table 12) and this variation was associated with cultivars and the age and class of birds. In general, the energy value of field peas is higher compared to those of faba beans and lupins, due mainly to their high starch content.

#### 2.2.2. Amino Acid Digestibility

The AA digestibility values (Table 13) in field peas vary depending on the cultivar and, age and class of birds. Szczurek and Świątkiewicz [97] reported a higher standardised ileal AA digestibility in field peas for 28-day old broilers than for 14-day old broilers. In the same study, a higher digestibility was determined for a white-flowered field pea (cv. Tarchalska) than for a coloured-flowered cultivar (cv. Milwa). A similar cultivar effect was recently reported by Adekoya and Adeola [98] for standardised ileal AA digestibility in broilers fed three field pea cultivars (cv. DS-Admiral, Hampton and 4010).

**Table 11 animals-14-00619-t011:** Amino acid content (g/kg, dry matter basis) of field peas.

Amino Acid	References								
	[22]	[24] ^1^	[25]	[49]	[50]	[58]	[90] ^2^	[96] ^3^	[99]
Essential									
Arginine	22.0	10.9	24.3	25.2	22.0	16.1	21.1	12.4	21.1
Histidine	6.2	2.8	5.8	6.6	6.5	4.3	6.3	5.1	6.3
Isoleucine	10.2	5.1	9.4	10.2	9.7	9.9	9.4	7.3	10.9
Leucine	16.8	8.2	16.6	17.5	17.5	16.0	16.7	12.7	18.3
Lysine	17.0	8.3	14.2	17.1	17.3	14.8	17.3	14.0	18.8
Methionine	2.3	1.0	1.9	2.2	2.6	2.0	2.5	2.2	2.7
Phenylalanine	11.0	5.2	11.1	11.5	10.9	10.8	11.1	8.9	11.9
Threonine	9.0	4.1	8.5	9.6	9.0	9.2	8.6	7.5	10.2
Tryptophan	na	0.9	na	na	na	2.0	na	1.4	2.3
Valine	12.3	5.5	10.8	12.0	10.5	10.3	10.3	8.3	13.0
Non-essential									
Alanine	10.4	5.0	10.1	11.3	10.0	10.1	9.8	8.1	11.4
Aspartic acid	26.7	12.5	25.4	28.6	28.6	30.1	26.6	20.9	28.9
Cysteine	1.8	1.4	3.2	3.5	3.1	3.4	3.3	3.5	3.5
Glycine	10.3	4.7	10.1	10.9	10.4	9.7	9.9	7.7	11.1
Glutamic acid	39.7	22.3	38.4	41.3	39.6	39.3	37.3	29.3	45.1
Proline	8.4	4.6	9.7	na	9.6	9.7	9.3	7.8	10.4
Serine	11.8	5.2	11.1	12.9	10.3	12.8	10.0	8.2	12.1
Tyrosine	7.3	3.0	6.7	7.6	8.3	7.0	7.8	5.7	7.1

^1^ Average of 4 cultivars (Amino, Australian, Finale and Frilène); ^2^ Average of 4 cultivars (Santana, Miami, Courier and Rex); ^3^ cv. Alvesta.

#### 2.2.3. Feeding Trials

Several studies have demonstrated the value of field peas as a protein source in poultry diets. According to Sipsas et al. [3], poultry diets can contain up to 250 g/kg field peas with little risk of wet droppings. Similarly, an inclusion of 200–300 g/kg of field peas in the diets of broilers and layers has been reported by Perez-Maldonado et al. [25], Farrell et al. [47] and Castell et al. [100]. According to Anderson et al. [85], field peas can be fed at 200–300 and 400 g/kg in the diet for broilers and laying hens, respectively. Janocha et al. [101] recommended inclusion levels of 100–150 and 200–250 g/kg field peas for broiler starters and growers, respectively. Brenes et al. [102] found that the performance of broilers fed diets containing 480 g/kg of field peas was similar to those fed a maize–soy diet. However, the inclusion of 600 g/kg field peas has shown to depress the egg production, egg mass and feed efficiency in laying hens [103].

**Table 12 animals-14-00619-t012:** Apparent metabolisable energy (MJ/kg dry matter basis unless otherwise specified) of field peas.

Cultivar	Bird Class	AME	AME_n_	References
Finale	Broilers	-	11.56	[104]
Finale	Adult roosters	-	11.77	[104]
Frisson	Broilers	-	10.86	[104]
Frisson	Adult roosters	-	11.28	[104]
Impala	Broilers	-	10.13 #	[105]
Radley	Broilers	-	10.29 #	[105]
Sirius	Broilers	-	8.28 #	[105]
-	Poultry	11.50 #	-	[3]
Glenroy	Pullets	11.70 *	-	[25]
-	Broilers	-	10.2–11.3 *	[106]
-	Broilers	11.7	-	[89]
Santana	Broilers	10.78	10.16–12.30	[31,90]
Miami	Broilers	10.15	9.81	[90]
Courier	Broilers	10.39	9.71	[90]
Rex	Broilers	9.82	9.11	[90]
Sohvi	Broilers	12.2	-	[44]
Karita	Broilers	13.8	-	[44]
Tarachalska	Broilers	-	9.05 *	[107]

* As is basis. # Basis (dry matter or as is) is not reported.

**Table 13 animals-14-00619-t013:** Ileal amino acid digestibility (apparent ^1^/standardised ^2^) of field peas.

Amino Acid	References			
	[49] ^1^	[96] ^2,3^	[97] ^2,4^	[98] ^2,5^
Essential				
Arginine	0.83	0.89	0.89	0.92
Histidine	0.75	0.90	0.85	0.87
Isoleucine	0.71	0.82	0.80	0.74
Leucine	0.71	0.83	0.82	0.85
Lysine	0.83	0.91	0.87	0.90
Methionine	0.70	0.90	0.83	0.83
Phenylalanine	0.72	0.82	0.85	0.86
Threonine	0.69	0.87	0.81	0.85
Tryptophan	na	0.78	na	0.86
Valine	0.71	0.81	0.82	0.84
Non-essential				
Alanine	0.73	0.82	0.84	0.86
Aspartic acid	0.78	0.77	0.85	0.87
Cystine	0.66	0.70	0.76	0.81
Glutamic acid	0.80	0.89	0.88	0.91
Glycine	0.71	0.80	0.83	0.85
Proline	na	0.86	0.83	0.85
Serine	0.71	0.79	0.82	0.87
Tyrosine	0.72	0.82	0.86	0.87

^3^ Cultivar Alvesta; ^4^ Average of white- and coloured-flowered cultivars; ^5^ Average of 3 cultivars (DS-Admiral, Hampton and field peas 4010).

### 2.3. Faba Bean

There are two types of faba beans, namely major (broad bean), with an average seed weight of 800 mg, and minor (horse bean, tic bean) with an average seed weight of 550 mg [3]. Faba beans are mostly consumed in Mediterranean countries, China and Brazil. The breeding of new cultivars with tannin-free seeds and with low vicine and convicine contents has offered new perspectives for the feed use of faba beans [12].

The reports on the nutrient composition of faba beans are summarised in Table 14. The large variation in the nutritional composition of faba beans probably reflects differences in cultivar, environment, growing condition and year of harvest [108,109]. The seeds are good sources of protein and starch (237–349 and 371–447 g/kg DM, respectively; Table 14). According to Chavan et al. [110], the crude protein content of faba beans varies from 200 to 410 g/kg. Rubio et al. [108] reported that the mineral contents vary considerably between cultivars (light- vs. dark-seed-coat cultivars) and seed fractions (cotyledon vs. hull). Light-seed-coat cultivars tend to have lower mineral and phytate contents than those with a dark seed coat [108].

The AA composition of faba bean is presented in Table 15. Faba bean is a good source of essential AA, especially lysine (7.1–21.8 g/kg). Methionine and cysteine (0.8–2.8 and 1.4–5.8, g/kg respectively) are the limiting AAs.

#### 2.3.1. Apparent Metabolisable Energy

The AME and AME_n_ (nitrogen-corrected AME) values reported for faba beans range from 8.8–12.4 and 8.1–12.7 MJ/kg, respectively (Table 16), which are comparable to those in SBM (8.4–10.6 MJ/kg) [83]. The variation in AME values is attributed to differences in cultivar and experimental methodology. Of interest is that tannin-free cultivars of faba beans tended to have higher AME values than those containing tannin. Brufau et al. [111] reported the AME_n_ values of spring and winter cultivars of faba beans as 9.18 and 9.92 MJ/kg, respectively, using total collection method and as 8.56 and 8.62 MJ/kg using chromic oxide index method, respectively. The same study also reported a reduced AME (9.06 vs. 10.35 for the total collection method and 7.84 vs. 9.33 for the index method) in coloured-tannin cultivars when compared to tannin-free white cultivars. These findings are in agreement with those of Vilariño et al. [121] who reported a reduced AME_n_ in high-tannin cultivars of reconstituted faba beans when compared to low-tannin cultivars. The same study [121] also reported a negative effect of vicine and convicine on the AME_n_ of reconstituted faba beans. On the other hand, the inclusion of faba beans (80–240 g/kg) has been shown to increase the AME_n_ of diets when compared to the AME_n_ of the control diet [122].

#### 2.3.2. Amino Acid Digestibility

The ileal AA digestibility AAs in faba beans is generally lower compared to those reported for SBM [83,123]. However, as can be seen in Table 17, the digestibility of most AAs is moderately high. The digestibility is highest for arginine (0.81–0.91) and lowest for cysteine (0.47–0.77).

#### 2.3.3. Feeding Trials

Perez-Maldonado [25] studied the inclusion level of 250 g/kg of grain legumes (faba beans, chickpeas, sweet lupins and field peas) on the productive performance of laying hens over a period of 40 weeks and reported a reduced feed intake, hen-day egg production, egg weight and egg mass, and inferior feed conversion efficiency in birds fed faba bean (cv. Fiord) diets compared to those fed other grain legume-based diets. Alagawany et al. [124] studied five faba bean replacement levels (0, 25, 50, 75 and 100%) as a substitute for SBM for laying hens and reported that SBM can be replaced with faba beans at levels less than 50% in laying hen diets. In the same study, the intakes of feed, protein and AME were decreased as the level of faba bean increased and the egg laying rate, egg output and feed efficiency were the lowest in hens receiving diets at 75 and 100% substitution.

Farrell et al. [47] examined different inclusion levels of faba bean for broiler chickens and recommended an inclusion level of 200 g/kg in broiler diets. Similarly, Nalle et al. [75] observed that faba beans can be included up to 200 g/kg in broiler diets without any detrimental effects on performance. Koivunen et al. [122] studied four inclusion levels (0, 80, 160 and 240 g/kg) of faba beans for broilers and concluded that a 160 g/kg faba bean can be safely used in broiler diets. These findings are in agreement with the results of Gous [119] and Ivarsson and Wall [125] who did not find any adverse effect of pelleted broiler diets with 200–250 g/kg faba bean on the growth performance of broilers. In contrast, the same study also reported a reduced feed intake and body weight in broilers fed the mash diet with the same inclusion level of faba beans, which suggests that the optimum faba bean inclusion level depends on the feed form. It is evident that the negative effect of feeding faba beans on the growth performance of poultry in the early studies is due to the high concentration of antinutrients in faba beans. However, with the development of plant breeding techniques, there are cultivars with zero-tannin or low vicine and convicine concentrations [117,120,126]. It has been reported that inclusions of 150, 300, 400–450 g/kg of zero-tannin faba bean cultivars is possible for broiler starter, grower and finisher, respectively [120,126].

Time of planting and harvesting faba beans, especially in the tropics, may influence the seed quality and its digestibility and consequently the growth performance in poultry. Smit et al. [117] studied the effect of the early or late planting and harvesting of two zero-tannin cultivars (Snowbird and Snowdrop) and a low-vicine and -convicine cultivar (Fabelle) on the nutrient digestibility, and reported that late planting and harvesting increased the digestibility of gross energy, protein and AA when compared to early-planting and -harvesting cultivars, regardless of increased proportions of frost-damaged beans in the late-planting and -harvesting cultivars. However, a subsequent study [126] did not find any negative effect on the growth performance of broilers fed low-quality (frost-damaged or immature) faba beans (150, 300, 450 g/kg for broiler starter, grower and finisher, respectively) when compared to those fed the high-quality seeds.

### 2.4. Chickpeas

Chickpeas are grouped into two types, namely ‘Desi’ and Kabuli’ varieties, based on seed size, colour and the thickness and shape of the seed coat. Desi type chickpeas are of Indian origin whereas Kabuli chickpeas are of Mediterranean, North African and West Asian origins. According to Nalle [12], Desi types produce smaller seeds, generally 400 or more seeds per 100 g. The seeds have a thick, irregular-shaped seed coat which can range in colour from light tan to black. Kabuli varieties (also referred to as garbanzo beans) produce larger seeds that have a thin seed coat with colours that range from white to a pale cream-coloured tan.

The crude protein content of chickpeas is moderate, ranging between 182 and 270 g/kg DM as summarised in Table 18. The starch content ranges between 310 and 535 g/kg DM. There are differences between the two varieties, with Desi varieties containing less starch (364 vs. 411 g/kg [127]) and more fibre (90 vs. 60 g/kg [128]) than the Kabuli varieties. The lipids in chickpeas comprise mostly of polyunsaturated fatty acid, with linoleic and oleic acids as the primary constituents [127]. The moderate content of fat (42–156 g/kg) and high starch content make chickpeas a good source of available energy for poultry. Chickpea is richer in phosphorous and calcium when compared to other grain legumes.

The AA composition of chickpeas is presented in Table 19. Glutamic acid is found in the highest concentrations in chickpeas, followed by aspartic acid and arginine. Chickpeas is a good source of lysine, but deficient in methionine and cysteine. A tryptophan content of 1.8 g/kg (as received basis) was reported in chickpeas [140]. As suggested by Chiaiese et al. [147], the use of transgenic techniques would help overcome the deficiency of these limited AAs.

#### 2.4.1. Apparent Metabolisable Energy

Published data on the AME of chickpeas are scant. According to Feedipedia [55], the AME of Desi chickpeas was 12.7 MJ/kg DM. However, INRA feed tables [148] reported an AME_n_ of 14.5 MJ/kg DM for chickpea for broilers. Using the European table of energy values for poultry feedstuffs [149], Viveros et al. [128] estimated the AME of Kabuli and Desi chickpeas to be 12.6 and 10.5 MJ/kg DM, respectively. The lower AME of the Desi type was attributed to its higher fibre content compared to Kabuli types (90–112 vs. 33–60 g/kg) [128,150]. The AME of chickpeas for other poultry species has also been reported. The AME of chickpeas was determined to be 10.5 MJ/kg for laying hens [25], 14.8 MJ/kg DM for adult roosters [148] and 12.8 MJ/kg for broiler turkeys (cv. Burnas; [136]).

**Table 19 animals-14-00619-t019:** Amino acid content (g/kg, dry matter basis) of chickpeas.

Amino Acid	References							
	[25] ^1^	[49]	[128] ^2^	[136,137] ^3^	[138] ^4^	[139]	[145] ^5^	[151]
Essential								
Arginine	17.6	25.6	22.8	20.1	19.2	na	19.9	14.4
Histidine	5.1	6.9	7.9	na	6.5	6.2	5.4	4.4
Isoleucine	8.5	11.4	10.3	9.1	9.7	10.4	6.7	6.6
Leucine	14.9	18.3	18.5	19.3	17.6	17.4	16.3	12.0
Lysine	11.8	15.2	14.7	18.8	16.4	14.5	15.1	9.4
Methionine	2.6	3.0	3.2	na	1.7	1.9	3.1	Na
Phenylalanine	11.4	13.8	15.0	12.5	11.0	13.1	11.5	10.3
Threonine	7.3	8.8	10.0	10.2	8.0	8.8	8.2	8.3
Tryptophan	na	na	Na	na	3.0	1.6	na	na
Valine	8.9	11.5	10.5	9.6	10.7	10.2	7.4	8.8
Non-essential								
Alanine	8.2	10.2	10.2	14.1	na	na	na	6.8
Aspartic acid	22.0	26.8	26.3	29.0	na	na	na	15.7
Cysteine	3.3	3.5	Na	na	4.1	2.1	3.7	Na
Glycine	7.9	9.3	9.6	9.2	na	na	na	7.9
Glutamic acid	31.3	38.9	49.1	49.7	na	na	na	24.9
Proline	8.1	na	Na	na	6.4	na	na	12.3
Serine	10.2	13.2	13.0	12.5	na	na	na	9.4
Tyrosine	5.8	6.6	7.6	6.3	6.9	6.2	4.4	7.9

^1^ cv. Amethyst; ^2^ Average of Desi and Kabuli; ^3^ cv. Burnas; ^4^ cv. Serifos; ^5^ Average of 16 varieties (8 Desi and 8 Kabuli). Abbreviation: na—not available.

#### 2.4.2. Amino Acid Digestibility

Only one published report is available on the digestibility of the AAs of chickpeas. Ravindran et al. [49] reported that the apparent ileal digestibility coefficient of AAs ranged from 0.58 for cysteine to 0.84 for arginine (Table 20). The poor digestibility of cysteine is probably related to the low concentration (2.1–4.1 g/kg; Table 19) of this AA in chickpeas. Ravindran et al. [140] reported an ileal digestibility coefficient of 0.71 for tryptophan in chickpeas.

#### 2.4.3. Feeding Trials

Viveros et al. [128] demonstrated that the dietary inclusion of chickpea, varieties Kabuli (0, 150, 300 and 450 g/kg) and Desi (75 and 150 g/kg), linearly reduced the performance of growing chickens and increased the relative weight and length of the intestinal tract in 28-day old broilers. They also found that the inclusion of Kabuli chickpea resulted in the lower digestibility of starch and protein, intestinal enzyme (α-amylase and trypsin) activities and AME_n_ compared to those fed the control diet. However, the performance of birds was improved by the autoclaving of chickpeas. Farrell et al. [47] studied different inclusion levels (0, 120, 180, 240 and 360 g/kg) of grain legumes (field peas, chickpeas, faba beans and sweet lupins) in broilers and reported that overall, weight gain and feed conversion ratio (FCR) were inferior in broiler starters fed chickpeas (cv. Amethyst) compared to those fed field peas and faba beans. The birds fed the chickpea diets had lower digesta viscosity when compared to those fed the lupins, and the heaviest weight of the pancreas when compared to those fed other grain legumes. However, the growth performance was not influenced by different chickpea inclusions in broiler finishers. It was concluded that the maximum inclusion level of chickpeas in broiler starter and finisher diets was 100 g/kg. Similarly, Algam et al. [132] suggested an inclusion of 100 g/kg chickpeas for broilers. Christodoulou et al. [138] studied three chickpea inclusion levels (0, 120 and 240 g/kg) for broilers and reported that feeding the diet with 240 g/kg chickpeas adversely affected the performance and carcass yield of broiler chickens. However, the same study found a similar performance between the birds fed the diet with 0 and 120 g/kg chickpeas and recommended an inclusion of 120 g/kg chickpeas for broilers. A recent study reported a negative effect of 50% replacement of chickpeas (315–344 g/kg) for SBM on intestinal histomorphology and microbial populations in broilers [152]. The inclusion of raw chickpeas was observed to induce disturbances in metabolism by means of the shortening and thickening of intestinal villi and in intestinal structure in the same study. Nevertheless, an inclusion of 200 g/kg chickpea inclusion has been suggested by Bampidis and Christodoulou [153] and Ciurescu et al. [137].

Perez-Maldonado et al. [25] concluded from their experiments with laying hens that good production can be achieved when the inclusion rate of chickpeas was 250 g/kg. However, it was suggested that it is safer to use lower inclusion levels because of pancreatic enlargement in hens fed chickpea diets, possibly due to the presence of trypsin and chymotrypsin inhibitors. Feeding chickpeas for other poultry species has also been reported. According to Ciurescu et al. [136], young turkeys can be fed 240 g/kg chickpeas as an alternative protein source. Sengül and Calisar [134], did not find any negative effect of feeding 200 and 400 g/kg chickpea on the production performance of laying quails.

## 3. Antinutritional Factors in Grain Legumes

Plants produce a complex array of deleterious compounds to protect themselves against predation by herbivores, insects, pathogens, and microorganisms and to fight against adverse environmental factors. These compounds, which impact the optimum utilisation of nutrients and reduce their digestion, absorption and metabolism and may generate adverse health effects, are termed as antinutritional factors (ANFs; [154]). Most ANFs, through various mechanisms, increase the intestinal secretions of digestive enzymes and mucins leading to increased endogenous protein losses and lowered AA availability [155]. The major ANFs and toxic compounds present in plant food are protease inhibitors, α-amylase inhibitors, non-starch polysaccharides (NSPs), oligosaccharides, phytates, oxalates, tannins, polyphenols, alkaloids, phytolectins, gossypol, saponins, cyanogenic glycosides, compounds causing favism, lathyrogens, goitrogens, phytoestrogens, anti-vitamin factors, toxic proteins, and food allergens etc. A detailed discussion of the structure and modes of action of all these ANFs is beyond the scope of the current review and the readers are directed to authoritative compilations by Liener [154], Cheeke and Shull [156] and Dolan et al. [157].

The biological effects of ANFs range from a mild reduction in performance to death, depending on the specific ANF and concentration. Furthermore, different poultry species and age groups respond differently. For example, hens are reported to be more sensitive than chicks [158]. Also, each legume species or cultivar is likely to have variable levels of ANFs and may have different biological effects [67]. Reported ANFs in chickpeas are protease inhibitors, amylase inhibitors, oligosaccharides, polyphenols and phytolectins [153]. Faba beans contain oligosaccharides, phytate, tannins, vicine and convicine, protease inhibitors, and lectins [3,159,160]. Lupins contain alkaloids, protease inhibitors, saponins, phytate and oligosaccharides [3,161]. Field peas are relatively free of ANFs and therefore pre-processing steps are unnecessary prior to the inclusion in diets. Phytate is present in all grain legumes and tannins are located in dark-seeded grains. Of the compendium of ANFs mentioned above, only the protease inhibitors, lectins, phytic acid, tannins and NSPs are discussed in this review, because of their practical relevance.

### 3.1. Proteinaceous ANFs of Grain Legumes

The proteinaceous ANFs include protease (trypsin, chymotrypsin) and α-amylase inhibitors, phytolectins and lipoxygenase [162]. These ANFs, except phytolectins, are collectively referred to as enzyme inhibitors [163].

#### 3.1.1. Protease Inhibitors

Protease (trypsin and chymotrypsin) inhibitors are small protein molecules of wide distribution in the plant kingdom and they are common constituents of legume seeds. Protease inhibitors are considered as ANFs because of their ability to inhibit the activity of proteolytic enzymes. According to their molecule size, protease inhibitors are classified into two families namely, Kunitz with a molecular weight of 20 kDa and Bowman-Birk of approximately 8 kDa [12,164]. The former one is a “single headed” structure, consisting of about 180 AA residues and mostly active against trypsin, while the latter is “double headed”, containing about 80 AA residues including 7 disulphide bridges and is active against trypsin and chymotrypsin [165,166]. Fernandez et al. [167] showed that the most important interactions in the Bowman–Birk trypsin inhibition complex were salt bridges and hydrogen bonds, whereas in the Bowman–Birk chymotrypsin inhibition complex was hydrophobic.

In legume seeds, trypsin inhibitors can be found in different locations, depending on the legume type. Avilés-Gaxiola et al. [168] reported that >90% of trypsin inhibitor activity (TIA) of faba beans is located in the cotyledons, while in chickpeas, the TIA is distributed in the cotyledons (77–76%), embryonic axis (12–16%) and seed coat (9–11%). Published TIA values for raw legume seeds are presented in Table 21.

Liener and Kakade [169] showed that protease inhibitors can cause growth depression, and pancreatic hypertrophy and/or hyperplasia in rats and chickens when fed legumes containing high levels of these detrimental constituents. Kakade et al. [170] reported that protease inhibitor was responsible for 40% of growth depression and pancreatic enlargement in rats fed raw soybeans. The inhibition of proteases in the small intestine would stimulate, by feedback control mechanisms, the pancreatic enzyme secretions [171]. Since pancreatic enzymes are rich in sulphur AAs, this stimulation would cause a loss of methionine and cysteine for body tissue synthesis.

When trypsin is inhibited by active trypsin inhibitors, proteins are not digested properly and the AAs become less available [172]. Kakade et al. [173] reported that active trypsin inhibitors “lock in” an appreciable proportion of cysteine which is already limiting in legume seeds. Consequently, the sulphur-AA requirements of the animal are increased.

Protease inhibitors in legume seeds can be removed by different processing methods (see Section 4). Nalle et al. [89] reported that extrusion cooking of field peas at 140 °C and 22% moisture decreased the TIA from 0.23 to 0.19 TIU/mg. Khatab and Arntfield [174] reported that boiling, roasting, microwave cooking and autoclaving processes totally removed the TIA in grain legumes. Asao et al. [175] demonstrated that a Bowman–Birk-type proteinase inhibitor from faba bean was most stable in acidic and neutral pH (2–8), but it lost its activity upon heat treatment at 100 °C in alkaline pH (≥9).

#### 3.1.2. Lectins

Lectins (haemagglutinins or phytohaemagglutinins) are defined as carbohydrate-binding proteins (or glycoproteins) of non-immune origin that can recognise and bind simple or complex carbohydrates in a reversible and highly specific manner [176,177]. Legume lectins are subdivided into two groups, namely lectins with indistinguishable subunits (for example, *Phaseolus vulgaris* lectins) and those with several subunits [177,178]. The amount of lectin in grain legumes is higher than the lectins in other plants. Lectins in legume seeds are concentrated in the cotyledons and endosperm. Lectin activity in the protein fraction in legume seeds was reported to be variable, from 10–100 g/kg of the total seed proteins and sometimes up to 500 g/kg [179]. Table 21 presents the reported lectin content of raw legume seeds.

**Table 21 animals-14-00619-t021:** Proteinaceous antinutritional factors in legume seeds.

Legume	References	TIA (TIU/mg)	CIA (IU/mg)	α-AI (IU/g)	Lectin (HU/mg)
Chickpea	[163,180,181]	6.2–39	6.1–12	3.1–11	2.7–6.2
Field pea	[89,90,100,163,181,182]	0.2–10.8	0.7–10.2	2.8–80	5.1–15.1
Pigeon pea	[163,168]	4.8–31.3	2.1–3.6	23–34	25
Cowpea	[163,168]	3.4–67	1.6	1.4–90	40–640
Faba bean	[116,181,183]	0.4–7.2	1.1–3.6	19	5.5–49
Kidney bean	[181,183]	3.1–31	4.0–21	1370	75–89
Mung bean	[168,184]	1.8–16	-	-	27
Lupinus spp	[32,57,168]	0.1–4.3	-	-	-
Soybean	[168,181]	46–94	30	939	693
Lentil	[163,181]	3.6–7.6	3.5–4.7	0	11–50

Abbreviations: TIA—trypsin inhibitor activity; TIU—trypsin inhibitor units; CIA—chymotrypsin inhibitor activity; IU—inhibitor units; AI—amylase inhibitor; HU—hemagglutinin units.

Dietary lectins, as a first step, bind to the epithelial cells in the gut and elicit changes in cellular and body metabolisms. This binding of lectins to cell surface glycoproteins causes agglutination, mitosis and other biochemical changes in the cell. Different lectins have different levels of toxicity, but not all lectins are toxic [174]. For example, lectin (ricin) derived from castor bean (*Ricinus communis*) is toxic while lectins from common beans, peas, lentils etc. are relatively non-toxic [176]. The ingestion of 10 g/kg soybean lectin in a diet containing autoclaved-soybean protein has been demonstrated to inhibit the growth of rats by 26% [185]. Unlike other proteins, legume lectins are highly resistant to digestive breakdown and substantial quantities of ingested lectins may be recovered intact from the excreta of animals fed legume-based diets [171,176]. Besides the high degree of resistance to proteolysis, the ability of lectins to bind brush border cells can cause damage to microvillus membrane, the shedding of cells and decrease in the absorptive capacity of the small intestine [176]. According to Vasconcelos and Oliveira [186], lectins also interact with enzymes, lowering the availability of AAs and altering their metabolism. All these effects will lead to nutrient malabsorption, impaired immunological function, poor growth and even death.

Lectins are, in general, thermolabile; however, some are resistant to moderately high temperatures [187]. Therefore, lectins in grain legumes can be reduced by thermal and other non-thermal processing methods (see Section 4). However, there are limited data to prove that any of these methods completely remove lectins. The decrease in lectins during the cooking process was probably because of the degradation of hemagglutinins (protein) into their subunits or to other unknown conformational changes in their nature [188].

### 3.2. Non-Proteinaceous ANFs of Grain Legumes

Non-proteinaceous ANFs include tannins, phytic acid, NSPs, α-galactosides, phenolics, saponins, cyanogens and toxic AAs. Of these ANFs, only tannins, phytic acid and NSPs are discussed herein.

#### 3.2.1. Tannins

Tannins are water-soluble polyphenolic compounds of varying molecular masses that have the ability to react with proteins, polysaccharides and other macromolecules, to precipitate proteins from aqueous solutions, inhibit digestive enzymes and decrease the utilisation of vitamin and minerals [189,190]. Tannins are classified into three major classes, namely condensed tannins (proanthocyanidins), hydrolysable tannins and phlorotannins [191,192]. Condensed and hydrolysable tannins are found in terrestrial plants whereas phlorotannins found only in marine brown algae [191]. Condensed tannins are flavonoid monomer consisting of flavan-3-ol or flavan 3, 4-diol units without a sugar core [192]. Hydrolysable tannins are polymers of phenolic acids (mainly gallic or hexahydroxy diphenic acid) esterified to a core molecule, commonly D-glucose. The phlorotannins are structurally less complex than hydrolysable and condensed tannins and formed through the polymerisation of phloroglucinol (1,3,5-trihydroxybenzene). Tannins in legumes seeds primarily belong to hydrolysable and condensed tannins [193]. Hydrolysable tannins are subject to breakdown by hydrolysis in the digestive tract, which results in gallic acid that is readily absorbed and excreted in the urine [194].

Tannins are found in all dark-coated legume seeds [195]. The content of tannins in legume seeds varies depending on factors such as cultivars and environmental conditions during growth and post-harvest storage [193]. Tannins are mostly located in the hull and are present mostly as condensed tannins [196]. Total tannin content in legume seeds ranges from 0.06 to 33 mg/g, with faba beans having higher tannin contents (Table 22). Avola et al. [197] examined 15 accessions of Sicilian faba beans and reported an average tannin content of 26 mg/g.

Tannins are considered as ANFs in monogastric nutrition with undesirable effects on feed intake, nutrient digestibility and productive performance [194,198]. Condensed tannins are known to inhibit several digestive enzymes, including amylases, cellulases, pectinases, lipases and proteases [199,200]. It has been well documented that feeding poultry with diets containing tannins depressed the performance of birds [201,202] through adverse effects on nutrient digestibility [200,203,204], increased endogenous AA flow [189], damage to the mucosal lining of the digestive tract [202] and impaired immune function [205]. The minimum threshold of dietary tannin content needed to elicit a negative growth response in poultry is not known [206].

The effects of tannins on the digestion of protein, AAs, fat and starch in monogastric animals have been investigated by several researchers [195,200,203,207]. The in vitro digestibility of protein in faba beans are known to be negatively influenced by the tannin content of the seeds [3]. Ortiz et al. [203] reported a significant negative correlation between the dietary condensed tannin level and AA digestibility. The decrease in AA digestibility in diets containing tannins was attributed to the binding of dietary tannins and dietary proteins and the complexation of tannins with digestive enzymes [196,208]. The enzyme activity (lipase and α-amylase) has also been shown to be reduced by high dietary tannins [200].

However, recent reports indicate that tannins, at low concentrations, benefit poultry in terms of improved intestinal microflora and gut morphology [190,191,209]. Xu et al. [210] found that dietary tannin contents of 50–75 mg/100 g improved the immunity and intestinal health of broilers challenged with necrotic enteritis.

#### 3.2.2. Phytic Acid

Phytic acid [*myo*-inositol 1,2,3,4,5,6-hexakis dihydrogen phosphate], which is the major storage form of P in seeds [211], comprising more than two-thirds of total P [212,213]. Phytic acid is present in seeds as a mixed salt, phytate, mainly involving divalent cations such as Ca^+2^, Mg^2+^, Zn^+2^, Fe^2+^, Cu^2+^ and Mn^2+^ resulting in the reduced solubility, absorption and availability of these cations in the small intestine of animals [214]. The amount of phytate in legume seeds is highly variable depending on the cultivar, growing conditions, harvesting techniques and methods of processing (Table 22).

**Table 22 animals-14-00619-t022:** Concentrations of tannin and phytate in grain legumes.

Legume	Tannins (mg/g)	References	Phytate (mg/g)	References
Chickpeas	0.8–4.9	[215,216]	1.21–12.3	[146,193,216]
Field peas	0.04–30.9	[174,193,217]	1.3–13.0	[93,100,193]
Faba beans	0.06–33.0	[119,188,197,215,217]	8.4–8.6	[188]
Lupinus spp	0.61–9.8	[217,218,219]	2.5–16.5	[218,219]
Soybean	0.39–0.45	[193,217]	6.2–41.3	[193]

Several comprehensive reviews are available on the role of phytate in poultry nutrition [213,214,220,221,222]. Animal nutritionists have long regarded phytate as both an indigestible nutrient and an ANF for monogastric animals. Because poultry possess insufficient inherent phytase enzyme activity, P bound in the phytate molecule is only partially available. Moreover, phytate is polyanionic with the potential to chelate positively charged nutrients, which is fundamental to the antinutritive properties of phytate. Recent research has demonstrated that phytate also compromises the utilisation of protein/AAs, starch, lipids, energy, calcium and trace minerals.

The adverse effect of phytic acid in reducing the availability of protein/AAs through the phytate–protein complex has been examined previously [223,224,225]. The reduced solubility of proteins because of protein–phytate complex can adversely affect certain functional properties of proteins which are dependent on their hydration and solubility, such as hydrodynamic properties (viscosity, gelation, etc.), emulsifying capacity, foaming and foam performance and dispersibility in aqueous media. Furthermore, phytate interactions with proteins are pH-dependent [226]. At pH values below the isoelectric point of the protein, the anionic phosphate groups of phytate bind strongly to the cationic groups of the protein to form insoluble complexes that dissolve only below pH 3.5 [227]. It has also been demonstrated that the ingestion of phytic acid can increase endogenous AA losses, thereby negatively influencing true AA digestibility [228,229]. The loss of endogenous protein from the ileum has a direct effect on the digestible energy value of the diet, depending on the AA composition of the protein leaving the terminal ileum. This direct energetic cost has been estimated to be as much as 24 kcal/kg DM intake for every 1 g/kg dietary phytic acid [228].

Phytate also has a detrimental effect on starch digestibility through several mechanisms [223,224]. Phytate may inhibit amylase activity either directly or via the chelation of calcium, which is a requisite cofactor for amylase. Additionally, phytate may interact with starch directly or indirectly by binding proteins that are closely associated with starch granules [223]. Phytic acid may decrease energy digestibility by reducing the digestibility of energy generating nutrients such as carbohydrates, lipids and protein [224]. Phytate has been reported to reduce the activity of pancreatic lipase in the small intestine of broilers [224], implying that the reduced fat digestibility by phytic acid could partly be a result of reduced activity of lipase. Phytic acid could also reduce fat digestibility and absorption by binding bile acids via divalent cations such as calcium to form insoluble phytic acid–mineral–bile acids complexes, thereby reducing fat digestion and absorption, and bile acid re-absorption [224].

#### 3.2.3. Non-Starch Polysaccharides

Non-starch polysaccharides are a large group of polysaccharide molecules excluding α-glucans and starch [230]. The NSPs are the main components of plant cell walls, including cellulose, hemicellulose and pectin, and a major part of dietary fibre [231]. The NSPs can be soluble or insoluble in aqueous media [232]. It is difficult to present a general description of the plant polysaccharides, partly because they are heterogenous complex compounds and partly because they have been classified in array of ways, depending on the interests of the investigators. However, many investigators, for analytical purposes, divide the NSPs into soluble and insoluble, based on solubility or extractability. It is generally accepted that the adverse effects of soluble NSPs are primarily associated with the viscous nature of the soluble polysaccharides, and their resultant effects on gastrointestinal physiology and morphology, and their interaction with gut microflora. The other modes of action include altered intestinal transit time, and changes in hormonal regulation due to a varied rate of nutrient absorption [230].

The solubility, concentration and molecular weight of NSPs play an important role in the viscosity. The soluble fraction of NSPs enhances the digesta viscosity by directly binding the water molecules at low concentrations or by interacting themselves to form a network as the concentration increases [233]. Water-holding capacity is another characteristic of NSPs that may influence the antinutritional properties of NSPs [234]. The ability to absorb large amounts of water and maintain normal motility of the gut becomes an important attribute of insoluble NSPs in poultry nutrition [235]. Choct [236] reported that insoluble NSPs affect not only the digesta transit time and gut motility, but also act as a physical barrier leading to lowered nutrient digestion. The increase in gut viscosity lowers the mixing of digestive enzymes and substrates in the intestinal lumen [230]. Combined with increased mucus production, NSPs can also increase the resistance of the unstirred water layer at the intestinal surface [234]. Furthermore, NSPs in cell walls physically inhibit the access of digestive enzymes to nutrients that are encapsulated within cell walls. Soluble NSPs, in particular, may stimulate microbial growth and increase the amount of microbial protein and fat at the terminal ileum. Certain NSPs may also stimulate the growth of toxin producing microbes, which may affect gut health and digestive function [234]. In addition, endogenous secretions, such as bile acids, may be bound by the viscous NSPs and consequently reducing the extent of recycling. All the above could eventually lead to a reduction in the digestion and utilisation of nutrients.

Legume NSPs are more complex in structure than those in cereals, containing a mixture of colloidal polysaccharides called pectic substances, namely galacturonans, galactan and arabinans [236]. Pectic substances are mainly found in the cotyledone of legume seeds, while, cellulose and xylans, which are the major NSPs in cereal grains, are only found in the hulls of most legume seeds [230]. Periago et al. [237] reported that the major constituents of total NSPs of chickpeas were cellulose, arabinose and uronic acids. The concentration of soluble, insoluble and total NSPs of grain legumes are summarised in Table 23.

Among the four grain legumes considered in this overview, Australian sweet lupins contain the highest total NSP content, followed by white lupins and chickpeas contain the lowest. Gdala [244] reported that the NSP content of lupins (320–400 g/kg) was higher than that of faba beans (177 g/kg) and field peas (185 g/kg) and was higher in sweet lupins than in white and yellow lupins. The main NSP sugar residues in lupins are glucose and galactose while the NSPs in field peas and faba beans are glucose, arabinose and uronic acids [244]. The main components of NSPs in field peas and faba beans are glucose (82 and 94 g/kg, respectively), arabinose (33 and 34 g/kg, respectively), xylose (13 and 27 g/kg, respectively), uronic acids (26 g/kg and 31 g/kg, respectively) and galactose (15 and 18 g/kg, respectively), as reported by Gdala and Buraczewska [114]. In chickpeas, 62% of the total NSP is in the seed coat [245]. Wood et al. [245] reported that there was a difference in the sugar residues of NSPs between different chickpea genotypes, where glucose was the most abundant residue in the Desi type, followed by arabinose, while arabinose was the abundant residue in the Kabuli type, followed by glucose.

## 4. Improving the Feeding Value of Grain Legumes

The efforts to inactivate or destroy ANFs present in grain legumes have been attempted through several strategies. The maximum or complete destruction of ANFs may require different processing treatments because of differences in physical and biochemical properties of ANFs [246]. On the other hand, severe processing can have a negative effect on the nutritional value of legumes. Processing can involve physical, chemical, thermal and bacterial means.

### 4.1. Physical Processing of Grain Legumes

#### 4.1.1. Dehulling

Dehulling is the most commonly used and effective method to reduce the deleterious effects of ANFs such as tannins and NSPs, with the remaining kernel having higher energy and protein contents [12]. Dehulling is the removal of the seed coat of grain legumes, which is one of the primary post-harvest processes of grain legumes to improve palatability, appearance, texture, cooking quality and digestibility [247]. The removal of hulls from legume seeds is accomplished with mechanically with attrition-type dehullers, roller mills or abrasive-type dehullers [247]. Attrition-type dehullers and roller mills are particularly suitable for dehulling and splitting legume grains with loose seed coats (field peas, faba bean), whereas abrasive-type dehullers are suitable for dehulling grains with more tightly adhering seed coats such as cowpeas and pigeon pea [247].

The beneficial effects of dehulling on the nutritional value of faba beans [183,188,248], lupins [249,250,251] and field peas [105,252,253] are well documented [24,31,254]. Crude protein and crude fat contents have been reported to increase by about 10–35% and 11–20%, respectively, after dehulling (Table 24). Furthermore, total NSPs considerably decreased (by 43–52%) in dehulled faba bean, lupin and field peas. The AA concentration of faba bean, Australian sweet lupins and field peas was also increased after dehulling.

Luo and Xie [188] reported that dehulling decreased the tannin content of faba beans, but increased the phytate content and trypsin inhibitor activity due to the lower concentration of these ANFs in the hulls compared to the cotyledons. The removal of the hull, which contributes a substantial portion to the total seed weight, caused the relative increase in the ANFs. Mariscal-Landin et al. [24] reported a 40% decrease in the tannin content of faba beans as a result of dehulling.

Dehulling has been shown to markedly improve the digestibility of DM [249], protein [105,249], AAs [251], starch [31,255], and AME [14,102] of grain legumes. It had been found that dehulling increased the AME by 18 and 30% for lupins and field peas, respectively, while increasing the apparent protein digestibility by 7 and 16%, respectively [56,102]. Breytenbach and Ciacciariello [256] reported that dehulling increased the AMEn value of Australian sweet lupins from 8.61 MJ/kg to 8.81 MJ/kg. In a study with broilers, Igbasan and Guenter [105] found that the improvement of AME of field peas by dehulling varied depending on the cultivar; brown-seeded field peas (cv. Sirius) had the highest improvement (24%), followed by green-seeded field peas (cv. Radley, 5%), and yellow-seeded field peas (cv. Impala, 3%). The observed improvements in AME were attributed to increases in starch digestibility as a result of the removal of indigestible fibre components and tannins in the hulls.

Dehulling has also been shown to improve the performance of poultry. Brenes et al. [249] showed that the dehulling of white lupins (cv. Amiga) improved the growth performance in broilers. Similarly, Farhoomand and Poure [253] found higher weight gain and improved feed conversion ratio in broilers fed a diet containing dehulled yellow field peas than those fed whole raw field peas. The beneficial effect of dehulling is reported to vary depending on the legume species. Olkowski et al. [79] found that dehulling markedly increased the weight gain in broilers fed diets containing narrow-leaf and white lupins, but not yellow lupins. However, the performance of broilers fed all lupin cultivars was still lower than those fed the control SBM diet. Igbasan and Guenter [103] demonstrated that feeding laying hens diets containing dehulled field peas improved egg production, feed intake, egg weight, yolk colour, albumen height and shell thickness. The positive impact on laying performance was due both to increases in the nutrient content and digestibility of the dehulled meal.

#### 4.1.2. Soaking

Traditionally, when used as human foods, grain legumes are soaked in water before cooking to improve cooking quality and reduce cooking time. This conventional method also reduces the ANFs due to their water solubility or the activation of degrading enzymes [163].

Soaking has been shown to reduce the concentration of phytate [183,252,257], tannins [183,257], polyphenols [183,257,258], trypsin, chymotrypsin and α-amylase inhibitors [183,259], lectins [260] and oligosaccharides [261] in grain legumes. The efficacy of soaking depends on the soaking solution [258], time [261], pH [262] and temperature of the soaking solution and the species of grain legumes [262]. It has been reported that soaking grain legumes in citric acid and sodium bicarbonate rather than water is more effective in reducing the tannin content (by 63 and 68%, respectively). However, in the same study, soaking reduced the total polyphenols in red kidney beans soaked in sodium bicarbonate solution (51% reduction) being the most effective one when compared to citric acid and water (45 and 13%, respectively). The benefits of soaking may vary depending on the species of grain legumes. Alonso et al. [252] observed reduced enzyme inhibitors (trypsin, chymotrypsin and α-amylase inhibitors) in water-soaked field pea cultivars, Solara and Ballet, but not in cv. Renata. The same study reported a reduced phytate content in all pea cultivars with soaking.

Soaking is also found to increase the digestibility of nutrients in grain legumes. Alonso et al. [183] reported an increased in vitro digestibility of protein and starch in faba beans. An improvement of in vitro protein digestibility with soaking was reported for green faba beans, but not for white faba beans [188]. In contrast, Avanza et al. [257] did not find any improvement in the in vitro protein digestibility of cowpea cultivars.

Combination of soaking with other processing methods (cooking or thermal treatment) was shown to be more effective in reducing the ANFs than soaking alone [188,259]. Han and Baik [261] reported that soaking legumes (chickpeas, field peas and lentils) along with ultrasound for 3 h or high hydrostatic pressure for 1 h was more effective when compared to soaking alone for 3 h to reduce the oligosaccharide content.

### 4.2. Germination

Germination is the first step of a seed’s growth as a plant. During the germination process, the enzymatic system of seed is activated for the hydrolysis and mobilisation of nutrients for plant growth, which also reduces the ANFs [163,263]. For example, phytic acid is hydrolysed by endogenous phytase to release the bioavailable inorganic phosphorous and other phytate-bound minerals and nutrients. Alonso et al. [252] reported a reduction in phytic acid, tannins, polyphenols and enzyme inhibitors in germinated field peas when compared to the raw pea seeds. The same study also reported the influence of germination time on the reduction in ANFs where the reductions were greater with increasing germination times (24, 48 and 72 h). Eskin and Wiebe [264] reported a 71–77% reduction in phytate content in two faba bean cultivars (Ackerperle and Diana) after germination. Khalil et al. [265] reported a reduction in phytate in chickpeas as a result of germination where the reduction was more pronounced in the Kabuli type (73%) than Desi type (32%). Yasmin et al. [258] reported a 43% reduction in phytate following germination of red kidney bean. However, Ferruzzi et al. [248] did not find any effect of germination on the tannin content of faba beans. According to Savelkoul et al. [266], germination results in a loss of biomass of more than 1% per day of germination. Further studies are necessary to confirm that the advantage of ANFs degradation outweighs the negative effect of biomass loss.

The germination process is also found to improve the nutrient content of grain legumes [263,265,267]. Dagnia et al. [263] reported that germination increased the protein content of sweet lupins from 395 to 435 g/kg DM), but the protein quality was lowered in germinated lupins. Khalil et al. [265] reported that germination increased the contents of protein, fat and ascorbic acid and in vitro protein digestibility in the Desi and Kabuli varieties of chickpea. Germination time [267,268] and grain legume species [265,267] also play a role in the influence of germination on ANFs.

### 4.3. Fermentation

Fermentation is a biochemical process used to enhance the bioavailability of nutrients, improve the organoleptic properties and extend the shelf life of plant seeds [163,269]. The process can be natural or artificial by the addition of microorganisms namely, bacteria (*lactobacillus*, *Bacillus*, *Streptococcus*, *Enterococcus*, *Aspergillus* etc.) and yeast [268,270,271]. During the fermentation process, complex molecules are converted into simple molecules by means of endogenous enzymes in microorganisms. As indicated by Maleki and Razavi [268] and Adebo et al. [271], fermentation process increased the contents of crude protein, AAs and fat in field peas and chickpeas. The in vitro digestibility of protein in lupins has been shown to increase by fermentation process [271]. In addition to improving the nutrient digestibility, the fermentation eliminates the pathogenic microorganisms, reduces ANFs in grain legumes and improve the antioxidant properties [268,269]. Microbial culture, enzyme activity and environmental conditions are the important factors that determine the outcome of the fermentation process [268]. The duration, temperature and pH of fermentation process vary from 2 h to several weeks, 30 to 42 °C and 4 to 9, respectively [268].

### 4.4. Exogenous Enzymes

The use of exogenous enzymes has become a common practice in the feed industry due to its effectiveness and lower cost compared to dehulling and thermal processing. The benefits of enzyme supplementation in poultry diets are wide ranging [272,273,274,275]. The aims are to (i) destroy the ANFs; (ii) increase the availability of nutrients such as starch and proteins that are enclosed within fibre-rich cell walls and, therefore, not accessible to endogenous digestive enzymes; (iii) breakdown specific chemical bonds in raw materials that are not usually broken down by the animal’s own enzymes; (iv) supplement the enzymes produced by young animals where, because of the immaturity of their own digestive system, endogenous enzyme production may be inadequate; (v) reduce the variability in nutritive value between samples of a feedstuff, (vi) improve gut health; and (vii) decrease nutrient overload in the manure.

Four main types of enzymes are commonly used in poultry diets, which are NSP-degrading enzymes (xylanase and β-glucanase), protein-degrading enzymes (protease), starch-degrading enzymes (amylase) and phytate-degrading (phytase) enzymes [276]. It is worth noting that feed ingredients typically contain more than one ANF; as a result, the addition of multi-enzymes is more effective to improve nutrient digestibility. It must be also noted that different feed enzymes have different modes of action.

In several grain legumes, NSPs are a major ANF. The antinutritive property of NSPs is due mainly to their ability to increase the digesta viscosity and their physiochemical characteristics [233]; therefore, NSP-degrading enzymes have been used to reduce these negative effects of NSP. In NSP-degrading enzymes, there are three mechanisms, namely cell wall breakage, viscosity reduction and provision of fermentable sugars to beneficial bacteria in the small intestine [272,277]. Microbial phytase is the second enzyme used in grain legume diets to mitigate the negative effects of phytic acid by releasing phytate-bound P and other nutrients.

Enzyme addition has been known to reduce the tannin [102] and insoluble NSP [42,277] contents in grain legumes. Brenes et al. [102] demonstrated that the enzyme treatment was beneficial when added to diets with a high-tannin cultivar (Maple) of field peas, but not for a low-tannin (Trapper) cultivar. The addition of a multi-enzyme cocktail (carbohydrase, protease and α-galactosidase) to a diet containing 700 g/kg raw lupins was found to improve the weight gain and feed conversion ratio of broiler chicks by 18 and 10%, respectively [56]. Naveed et al. [278] reported that the addition of xylanase or cellulase in lupin-based diets improved the performance of broilers, probably due to the elimination of the adverse effects of cellulose and NSP. Cowieson et al. [279] reported that the negative effect of adding pea meal (300 g/kg) to broiler diets could be overcome by supplementing carbohydrases (α-amylase, pectinase and cellulase). However, Kocher et al. [280] did not find any improvement in broilers fed a combination of pectinase, protease and amylase enzymes, but found an increased AME_n_ in birds fed these enzymes in a diet low in energy and protein.

### 4.5. Thermal Processing of Grain Legumes

Thermal processing can be categorised into dry and wet heat treatments [12]. In thermic processes with water, the main effects are to inactivate heat-labile ANFs, such as protease inhibitors and lectins, and to increase nutrient digestibility, especially of AAs. Dry heat processes, on the other hand, will improve the acceptability and the nutritional components of feed ingredients. Roasting, popping and micronising are examples of dry heat treatment, while pelleting, expansion, extrusion, compacting and steam flaking represent the wet heat treatments [281].

The use of appropriate processing temperature is critical for the elimination of heat-labile ANFs found in legume seeds [12]. Under-processing will have negative effects on the digestibility of AAs since the ANFs are not fully eliminated. Excessive heat treatment, or over-processing, will also lower AA digestibility as AAs may be destroyed or become unavailable due to the formation of indigestible complexes. The amino acids that are most affected by over-processing are lysine and cysteine. Cysteine is the most heat-labile AA, while the effect on lysine may be explained by the Maillard reaction in which free lysine binds with free carbonyl groups of reducing sugars to form Maillard complexes. In advanced stages of the Maillard reaction, the AA becomes completely unavailable due to cross-linkages being formed between protein chains [282].

Several studies have shown that excessive thermal treatment reduces the digestibility of legume proteins for poultry as a result of protein aggregation [283,284,285,286,287]. Speed et al. [288] reported that increasing the temperature increases the rate of aggregation by increasing both frequencies of molecular collision and hydrophobic interaction. Temperature may also change the relative composition of secondary structures and alter the aggregation behaviour. Temperatures up to 70 °C usually affect most proteins reversibly or partially, while temperatures between 70 to 100 °C will break the hydrogen bonds, disulphide bonds, and alpha helix secondary structure. Mild heating (100 to 150 °C) damages tertiary protein structures with little effect on nutritional value. At 105 to 150 °C, the losses of lysine, serine and threonine become prominent while isopeptides such as lysinolysine and glutamyllysine are formed and cross-links between proteins are generated. The level of isopeptide formed, as well as cross-links between proteins, proportional to the degree and temperature of heating. Above 150 °C, the pyrolysis of AAs occurs, i.e., the destruction of the AA with a large number of potential end-products, some of which are carcinogenic [12].

Extrusion is a process wherein the feed is subjected to mixing, shearing and heating under high pressure before the extrudate finally is forced through a die [289]. Feed may undergo reactions during processing that could be beneficial if the nutritional value is improved or detrimental if nutrients are destroyed and/or become resistant to digestion. Reactions that occur in the feed during extrusion are largely determined by the shear force, temperature, moisture, resident time and pH [290]. The reactions also depend on the type of reactant present, such as water, lipids, carbohydrate and proteins. The functions performed by extrusion cooking include the gelatinisation of the starchy component, denaturation of proteins, stretching or restructuring of tactile components and the exothermic expansion of extruder and modification of liquids [12]. The primary aim of extrusion is to achieve a high level of starch gelatinisation and disruption of the grain structure. When the mass is cooked, the product is shaped by the die. The starch particles are expanded to form an open ‘honeycomb’-like structure, which is referred to as being ‘gelatinised’ [12]. During extrusion, proteins start to denature and convert from soluble to insoluble by the formation of bonds [257,291]. Some or all of these bonds are then broken by the increasing heat and shear to form a concentrated solution or melt phase that can produce formation of covalent bonds at high temperatures. Upon cooling, non-covalent and disulfide bonds form, and finally, if the moisture content is low enough, amorphous regions form that becomes crystalline.

Extrusion cooking may also affect the nutritional value of lipids as a result of oxidation, hydrogenation, isomerisation or polymerisation [292], and change the composition of starch and dietary fibre [293,294]. According to Lue et al. [295], the changes in dietary fibre profile after extrusion may be explained by three mechanisms. First, the starch forms fractions resistant to enzymatic attack, increasing the dietary fibre content. Second, the degradation of dietary fibre to low molecular weight fractions could lower the dietary fibre content. Third, the macromolecular degradation of fibre increases its solubility and changes its physiological effects.

The other benefits of extrusion process include decreasing ANFs, increasing digestibility of individual feed components, destruction of pathogens, and extension of feed storage time. Extrusion also lowers raw or bitter flavours commonly associated with many plant food sources. Many of these undesirable flavours are volatile in nature and are eliminated through the extrusion and decompression at the extruder die. Extrusion has been reported to have positive effects on the in vitro digestibility of protein [183] and digestibility of fat [296], AAs [296], and starch [58,183] of grain legumes. The improvement of protein digestibility after extrusion was likely to be due to the destruction of ANFs. In the case of starch digestibility, the improvement was probably due to changes in starch structure, such as fusion, gelatinisation, fragmentation and dextrinisation [12]. Extrusion was also known to alter the structure of protein (insolubilisation) in field peas and kidney beans [183,252]. According to Son and Ravindran [74], the extrusion had no effect on the digestibility of protein and AAs of white lupins while adversely affected the AME in broilers, which is in agreement with the results of Breytenback and Ciacciariello [256].

A reduction in the ANFs in grain legumes as a result of the extrusion process is evidenced in several research studies [58,183,252,297,298]. Alonso et al. [252] reported reductions in the contents of number of ANFs (condensed tannin, trypsin, chymotrypsin, α-amylase inhibitors and haemagglutinating activity) in field peas as a result of extrusion at 148 °C, 25% moisture and 100 rpm. van der Poel [297] demonstrated that extrusion (at 105–140 °C and 14–33% moisture) reduced the tannin content by 30–40% in two pea cultivars (Finale and varC306). Table 25 summarises the concentration of nutrients and selected ANFs of raw and extruded grain legumes.

Autoclaving, cooking, wet-heating, microwaving, roasting, expansion, flaking, high-pressure processing, micronisation or infrared radiation and jet soldering (hot air) are some other thermal processing methods [14,163,257,284,299]. Avanza et al. [257] demonstrated a reduction in polyphenols, tannin and phytic acid in cowpea cultivars by autoclaving (at 2.2 kPa pressure and 121 °C temperature) and cooking when compared to soaking. Dänicke et al. [284] reported a beneficial effect of jet sploder on the feeding value for broilers and laying hens when compared to hydrothermal and microniser processing methods. Tannin and phytic acid contents of faba bean have been shown to be reduced by low doses (0.5 and 1.0 kGy) of gamma irradiation [300]. Ferruzzi et al. [248] reported that flaking reduced the tannin content of faba beans by 20%.

### 4.6. Plant Breeding

Breeding has long been used in agriculture to modify the genetic make-up of plants to alter the contents of nutrients and ANFs [301]. The genetic changes which have occurred during the course of domestication of crops which have evolved into current grain legumes have been eloquently described by Smartt [302]. Because of the existence of genetic variation [66] in most ANFs, it is possible to achieve a substantial reduction in their concentration by plant breeding. Older cultivars of grain legumes are known to contain high concentrations of various ANFs, which severely limit their inclusion in animal nutrition. The levels of these constituents have been considerably reduced in modern cultivars through well-planned plant breeding technologies and this has enhanced the usefulness of grain legumes in animal feeding [303].

Examples of success stories include inter alia the elimination of vicine and convicine in faba beans [304] and the development of low-alkaloid lupins [305].

## 5. Concluding Remarks

Grain legumes are widely grown throughout the world and have been used as important sources of protein in human nutrition since ancient times. The botany, agronomy, processing and utilisation of these crops are well documented. Owing to their importance as human food, a wealth of data has been generated over the years on their nutrient composition. The current overview deals with the feeding value of four selected grain legumes, namely lupins, field peas, faba beans and chickpeas, in poultry feeding. The use of these legumes in poultry diets is driven by the shortage and cost of SBM, the conventional protein source for poultry. The reported variability in nutrient composition is dealt with in the review. In general, they are fairly moderate protein sources. The essential AA profile is high in lysine and low in methionine. The ileal AA digestibility is comparable to that of SBM. Because lysine is the first limiting AA in cereal-based poultry diets, grain legumes have a high complementary value. Grain legumes, with the exception of lupins, are also good sources of available energy.

The paucity of published information on the ileal digestibility of AAs and AME for poultry are highlighted in this review. Reliable data on the variability in ileal AA digestibility and AME are needed for precise feed formulation and future evaluations should focus on generating more data on these parameters. The list of ANFs found in grain legumes is formidable; fortuitously, however, the antinutritive effects could be effectively eliminated or substantially reduced by proper processing. A summary of nutritionally-relevant ANFs and legume processing methods is also provided in this paper. It is concluded that the four grain legumes reviewed herein, when adequately supplemented with methionine, have a feed value equal to that of SBM if the level of inclusion does not exceed 200–300 g/kg.

## Figures and Tables

**Table 1 animals-14-00619-t001:** Classification of types of grain legumes (adapted from FAO [2]; Mitchell et al. [4]).

Dry common bean (*Phaseolus vulgaris* L.) ^1^
Dry faba beans (*Vicia faba* L.) ^2^
Dry field peas (*Pisum sativum* L.)
Chickpeas (*Cicer arietinum* L.)
Lupins (*Lupinus* spp.) ^3^
Lentils (*Lens culinaris* L.)
Pigeon peas (*Cajanus cajan* L.)
Cowpeas (*Vigna unguiculata* L.)
Bambara bean (*Vigna subterranea* L.) ^4^
Vetches (*Vicia sativa* L.)
Pulses nes ^5^
- Green gram (*Vigna radiata* L.) - Black gram (*Vigna mungo* L.) - Horse gram (*Dolichos biflorus* L.) - Winged bean (*Psophocarpus tetragonolobus* L.) - Lima bean (*Phaseolus lunatus* L.) - Hyacinth bean (*Lablab purpureus* L.) - Lathyrus (*Lathyrus sativus* L.) ^6^ - Guar bean (*Cyamopsis tetragonoloba* L.) ^7^ - Moth bean (*Phaseolus acantifolius* L.) - Rice bean (*Vigna umbellata* L.) - Adzuki bean (*Vigna angularis* L.) - Velvet bean (*Mucuna pruiens* L.)

^1^ Includes pinto, black, and kidney beans and dry peas (e.g., yellow or green peas). ^2^ Also known as fava or broad beans. ^3^ With over 250 species, the taxonomy of lupins has always been confusing. The major species of interest as protein crops are *Lupinus albus* (white lupin), *Lupinus angustifolius* (blue lupin or narrow leafed lupin) and *Lupinus luteus* (yellow lupin). Low alkaloid lupins are referred to as sweet lupins (e.g., *Lupinus angustifolius* with low alkaloids—Australian sweet lupin). ^4^ Also known as Bambara groundnuts. ^5^ Minor pulses that do not fall into any of the other categories. ^6^ Grass pea or Indian pea. ^7^ Cluster bean.

**Table 3 animals-14-00619-t003:** Amino acid content (g/kg, dry matter basis) of Australian sweet lupins.

Amino Acids	References				
	[22]	[24]	[25] ^1^	[26] ^2^	[26] ^3^
Essential					
Arginine	31.5	11.7	34.4	29.9	31.7
Histidine	11.0	3.1	8.0	7.6	7.6
Isoleucine	13.8	5.2	12.6	11.4	11.4
Leucine	21.9	7.9	20.8	20.6	21.1
Lysine	15.0	5.1	12.9	13.8	14.2
Methionine	2.6	0.8	1.8	2.0	4.5
Phenylalanine	12.2	4.3	12.5	10.8	10.6
Threonine	11.6	3.7	10.9	10.0	10.2
Valine	13.8	4.7	12.2	11.2	11.2
Tryptophan	na	0.8	na	2.8	2.9
Non-essential					
Alanine	11.0	4.0	10.7	10.0	10.4
Aspartic acid	30.8	11.0	29.4	29.4	30.8
Cysteine	2.5	1.5	33	3.6	3.7
Glycine	13.4	4.6	12.9	12.1	12.6
Glutamic acid	64.2	26.8	56.0	65.1	65.6
Proline	11.7	4.8	13.2	na	na
Serine	16.4	5.7	15.2	14.4	14.1
Tyrosine	11.1	3.0	10.2	9.5	10.2

References: ^1^ cultivar Gungurru; ^2^ cultivar Warrah; ^3^ cultivar transgenic high-methionine lupin; na = not available.

**Table 4 animals-14-00619-t004:** The apparent metabolisable energy values (MJ/kg dry matter basis unless otherwise specified) of Australian sweet lupins.

Cultivar	AME	Nitrogen-Corrected AME (AMEn)	References
Unknown	9.99 *	9.85 *	[22]
Danja	6.50–10.50	-	[38,40]
Gungurru	6.53–11.64	-	[25,38,40,41,42]
Warrah	9.42	-	[26]
Transgenic lupin	10.18	-	[26]
Wallan	6.38	5.35–5.82	[31,32]
Tanjil	6.73	6.18	[32]
Borre	7.12	5.52	[32]
Boruta	-	9.27	[43]
Neptun	-	8.67	[43]
Sonet	-	9.16	[43]
Graf	^-^	7.91	[43]
Pershatvet	7.00	-	[44]
Kadryl	7.37–8.40	-	[45]
Regent	6.04–6.88	-	[45]
Dalbor	6.71–7.68	-	[45]
Bojar	8.52–9.25	-	[45]
Tango	7.60–7.74	-	[45]

* As is basis.

**Table 6 animals-14-00619-t006:** The nutritional composition (g/kg, dry matter basis) of white lupins.

Nutrient	Average	Range *	References
Dry matter	911	886–944	[13,24,27,30,54,56,57,58,59,60,61,62]
Crude protein	362	202–424	[13,24,27,30,35,54,56,57,58,59,60,61,62,63,64,65,66,67,68]
Crude fat	102	60–130	[13,27,30,35,54,56,57,58,59,60,61,62,63,64,65,66]
Crude fibre	134	105–162	[13,24,27,30,54,58,59,60,62,63,65,66]
Acid detergent fibre	158	130–172	[24,27,56,58,60,61,62]
Neutral detergent fibre	203	185–234	[24,27,56,58,60,61,62]
Total fibre		344–394	[64]
Soluble fibre	44	36–52	[64]
Insoluble fibre	325	308–342	[64]
Starch	50	14–125	[27,58,64,68]
Ash	38	27–46	[24,27,30,54,56,57,58,60,62,63,64,65,68]
Calcium	2.3	1.6–3.2	[13,27,56,59,61,62]
Phosphorus	4.1	3.3–5.2	[13,27,56,59,61,62]

* Range is based on the average values reported in the given references.

**Table 7 animals-14-00619-t007:** Amino acid content (g/kg, dry matter basis) of white lupins.

Amino Acid	References							
	[24] ^1^	[57] ^2^	[58] ^3^	[59] ^4^	[67]	[70]	[71] ^5^	[72] ^6^
Essential								
Arginine	11.4	36.3	38.4	28.0	43.1	29.9	35.8	38.6
Histidine	2.5	8.8	9.0	7.0	9.4	7.1	5.9	8.3
Isoleucine	5.3	13.4	17.9	14.0	18.0	15.2	17.1	14.3
Leucine	8.3	26.0	28.6	25.7	28.7	23.3	23.4	24.3
Lysine	5.1	16.7	16.4	16.2	19.3	15.9	17.4	16.4
Methionine	0.7	2.8	2.6	6.5	na	3.4	2.9	2.6
Phenylalanine	4.1	13.1	16.1	14.6	na	11.9	13.7	12.4
Threonine	4.0	13.7	14.3	13.1	14.8	8.0	14.7	11.6
Valine	4.9	13.7	15.1	13.8	17.2	15.0	10.6	14.5
Tryptophan	0.8	na	2.3	3.2	3.2	na	3.4	na
Non-essential								
Alanine	3.7	12.0	12.7	na	na	10.9	na	10.2
Aspartic acid	11.6	34.4	45.7	na	na	33.8	na	33.6
Cysteine	1.4	5.3	5.5	7.5	na	na	3.7	5.1
Glycine	4.2	12.7	14.9	13.4	na	12.8	na	13.4
Glutamic acid	25.6	64.7	88.6	na	na	62.6	na	58.6
Proline	3.9	11.9	16.5	na	na	na	na	12.8
Serine	5.7	15.0	23.9	na	na	8.8	na	14.6
Tyrosine	5.1	13.8	17.6	na	na	9.8	na	13.4

^1^ cv. Lublanc; ^2^ average of three cultivars (Promore, Kiev mutant and Ultra); ^3^ cv. Multitalia; ^4^ cv. Buttercup; ^5^ cv. Hanti; ^6^ cv. Amiga; na = not available.

**Table 8 animals-14-00619-t008:** Apparent metabolisable energy (MJ/kg, dry matter basis) of white lupins.

Cultivar	AME	Class of Poultry	References
Amiga (alkaloid-free)	9.90	Broilers	[56]
Ultra	9.20	Roosters	[51]
Kiev mutant	9.58–13.29	Broilers	[38,40,42,57,74]
Promore	9.68	Broilers	[57]
Ultra	8.05	Broilers	[57]

**Table 9 animals-14-00619-t009:** Apparent ileal amino acid digestibility coefficient of white lupins for broilers.

Amino Acid	References		
	[49]	[57] ^1^	[74] ^2^
Essential			
Arginine	0.88	0.95	0.97
Histidine	0.81	0.81	0.82
Isoleucine	0.77	0.88	0.86
Leucine	0.79	0.89	0.88
Lysine	0.81	0.90	0.90
Methionine	0.84	0.83	0.79
Phenylalanine	0.79	0.92	0.92
Threonine	0.75	0.84	0.80
Valine	0.75	0.85	0.86
Tryptophan	na	na	na
Non-essential			
Alanine	0.78	0.85	0.84
Aspartic acid	0.80	0.87	0.78
Cysteine	0.83	0.81	0.84
Glycine	0.79	0.86	0.87
Glutamic acid	0.85	0.93	0.84
Proline	na	0.85	0.85
Serine	0.78	0.85	0.87
Tyrosine	0.81	0.88	0.88

^1^ Average of 3 cultivars (Promore, Kiev mutant and Ultra); ^2^ cv. Kiev mutant; na: not available.

**Table 14 animals-14-00619-t014:** Nutrient composition (g/kg, dry matter basis) of faba beans.

Nutrient	Mean	Range *	References
Crude protein	291	237–349	[3,24,25,33,44,58,111,112,113,114,115,116,117,118,119,120]
Crude fat	16	10–28	[3,25,33,44,58,112,113,114,115,116,117,118,119,120]
Crude fibre	106	84–232	[3,24,33,44,58,111,112,113,114,117,119,120]
Acid detergent fibre	116	83–133	[3,24,25,58,112,114,115,117,118,120]
Neutral detergent fibre	178	126–313	[3,24,25,58,112,114,117,118,120]
Ash	36	28–52	[3,24,25,33,44,58,111,114,115,116,117,119,120]
Starch	412	371–447	[58,107,112,114,116,117,118,120]
Calcium	1.3	1.0–1.7	[3,113,115,119,120]
Phosphorus	4.8	4.2–5.6	[3,113,115,120]

* Range is based on the average values reported in the given references.

**Table 15 animals-14-00619-t015:** Amino acid content (g/kg, dry matter basis) of faba beans.

Amino Acid	References							
	[24] ^1^	[25] ^2^	[44] ^3^	[58]	[116] ^4^	[117] ^5^	[119] ^6^	[120] ^7^
Essential								
Arginine	9.8	26.5	27.8	26.2	23.8	24.4	27.9	25.4
Histidine	3.2	6.7	8.2	7.1	6.6	7.2	Na	8.0
Isoleucine	4.8	10.8	11.7	12.6	9.2	12.7	11.0	11.8
Leucine	8.3	19.2	21.3	21.3	16.7	21.6	21.0	21.2
Lysine	7.1	14.4	18.6	18.0	14.0	18.8	21.8	17.0
Methionine	0.8	1.7	2.2	2.4	2.2	2.2	2.4	2.8
Phenylalanine	4.6	11.3	12.7	12.3	9.3	12.8	na	12.8
Threonine	3.8	9.4	10.4	10.2	7.5	9.7	6.7	9.1
Valine	5.4	12.1	13.6	13.8	10.4	13.7	13.0	13.5
Tryptophan	0.8	na	na	2.6	Na	2.3	2.5	3.2
Non-essential								
Alanine	4.5	10.9	12.5	12.0	10.1	13.0	na	11.3
Aspartic	11.9	27.5	24.3	28.0	26.1	30.3	na	30.6
Cysteine	1.4	3.0	4.1	3.7	3.6	3.2	3.9	5.8
Glycine	4.7	11.0	12.6	12.0	9.7	12.0	na	13.1
Glutamic acid	20.7	40.7	47.1	48.6	38.2	47.5	na	45.5
Proline	4.4	11.2	12.9	13.4	8.3	12.2	na	13.1
Serine	5.3	12.7	14.1	14.9	8.9	12.0	na	12.6
Tyrosine	3.3	7.8	10.0	8.5	7.5	9.3	na	10.1

^1^ cv. Alfred; ^2^ cv. Fiord; ^3^ Average of two cvs. (Kontu and Ukko); ^4^ Average of four cvs. (PGG Tic, Spec Tic, South Tic and Broad); ^5^ Average of early and late harvested beans of three cvs. (zero-tannin cvs: Snowbird and Snowdrop, and low vicine and convicine cultivar: Fabelle); ^6^ cv. Fiord; ^7^ Average of three zero-tannin cvs. (Snowbird, Snowdrop and Tabasco). na: not available.

**Table 16 animals-14-00619-t016:** Apparent metabolisable energy (MJ/kg dry matter basis unless otherwise specified) of faba beans for broilers.

Cultivar	AME	AMEn	References
Spring	-	9.2	[111]
Winter	-	9.9	[111]
Diana	-	8.9	[111]
Fiord	11.0–11.3 *	-	[25,119]
-	-	9.5–10.8 *	[106]
Reconsitituted beans ^1^	-	11.8–12.7	[121]
PGG Tic	10.8	9.8–10.5	[31,116]
Spec Tic	9.2	8.3	[116]
South Tic	12.0	10.6	[116]
Broad	8.8	8.5	[116]
Merlin	-	11.6 #	[118]
Olga	-	10.1 #	[118]
Albus	-	8.1 #	[118]
Amulet	-	7.9 *–12.2 #	[107,118]
Kasztelan	-	11.9 #	[118]
Kontu	12.4		[44]
Ukko	11.9	-	[44]

* As is basis. # Basis (dry matter or as is) is not reported. ^1^ Hulls and cotyledons of different faba bean cultivars (Gloria, Divine and Meli) were mixed in different ratios.

**Table 17 animals-14-00619-t017:** Apparent ^1^/standardised ^2^ ileal digestibility coefficient of amino acids in faba bean for broilers.

Amino Acid	[31] ^1^	[44] ^1,3^	[49] ^1^	[116] ^1,4^	[117] ^2,5^	[118] ^1^
Essential						
Arginine	0.91	0.90	0.81	0.90	0.88	0.91
Histidine	0.70	0.82	0.72	0.72	0.79	0.85
Isoleucine	0.85	0.82	0.68	0.83	0.77	0.84
Leucine	0.85	0.85	0.70	0.84	0.80	0.84
Lysine	0.91	0.88	0.76	0.89	0.83	0.90
Methionine	0.86	0.75	0.63	0.81	0.63	0.90
Phenylalanine	0.86	0.80	0.72	0.88	0.80	0.85
Threonine	0.84	0.79	0.68	0.77	0.72	0.81
Tryptophan	na	na	na	na	0.80	na
Valine	0.83	0.85	0.68	0.81	0.75	0.85
Non-essential						
Alanine	0.89	0.86	0.71	0.86	0.80	0.86
Aspartic acid	0.86	0.84	0.71	0.87	0.80	0.86
Cysteine	0.63	0.49	0.58	0.56	0.47	0.77
Glycine	0.81	0.77	0.67	0.76	0.65	0.82
Glutamic acid	0.90	0.87	0.75	0.88	0.87	0.90
Proline	0.71	0.75	na	0.54	0.75	0.83
Serine	0.86	0.81	0.69	0.79	0.78	0.85
Tyrosine	0.84	0.76	0.70	0.80	0.77	0.81

^3^ Average of two cultivars (Kontu and Ukko). ^4^ Average of four low-tannin cultivars (PGG Tic, Spec Tic, South Tic and Broad). ^5^ Average of early- and late-harvested beans of three cultivars (Zero-tannin cultivars: Snowbird and Snowdrop, and low vicine and convicine cultivar: Fabelle).

**Table 18 animals-14-00619-t018:** Nutrient composition (g/kg, dry mater basis) of chickpeas.

Nutrient	Mean	Range *	References
Dry matter	905	882–935	[3,25,128,129,130,131,132,133,134,135,136,137,138,139,140]
Crude protein	225	182–270	[3,25,128,129,130,131,133,136,137,138,139,140,141,142,143,144,145,146]
Crude fat	58	42–156	[3,25,128,130,131,133,136,137,138,139,141,142,144,145,146]
Crude fibre	79	42–75	[3,128,130,131,133,136,137,138,139,146]
Acid detergent fibre	93	45–115	[3,25,131,136,139,144,145]
Neutral detergent fibre	187	141–247	[3,25,131,136,139,144,145]
Soluble fibre	43	43	[141]
Insoluble fibre	235	235	[141]
Ash	37	29–60	[3,25,128,130,133,136,137,138,139,141,143,145,146]
Starch	422	310–535	[128,131,142,144]
Calcium	2.4	1.4–4.8	[3,128,133,145]
Phosphorus	4.0	3.9–4.1	[3,128,133,145]

* Range is based on the average values reported in the given references.

**Table 20 animals-14-00619-t020:** Apparent ileal amino acid digestibility coefficients in chickpeas for broilers.

Amino Acid	Digestibility Coefficient
Essential	
Arginine	0.84
Histidine	0.77
Isoleucine	0.70
Leucine	0.70
Lysine	0.76
Methionine	0.72
Phenylalanine	0.78
Threonine	0.70
Valine	0.73
Non-essential	
Alanine	0.73
Aspartic acid	0.73
Cysteine	0.58
Glycine	0.68
Glutamic acid	0.78
Serine	0.74
Tyrosine	0.72

Source: [49].

**Table 23 animals-14-00619-t023:** Soluble, insoluble and total non-starch polysaccharides (NSPs) contents of some grain legumes and soybean meal (g/kg dry matter).

Grain Legume	Soluble NSPs	Insoluble NSPs	Total NSPs	References
Chickpea	20–33	74–76	96–107	[233,237]
Faba bean	17–22	182–227	190–243	[31,114,116,238]
Australian sweet lupin	22–40	229–464	251–496	[31,32,233,239]
White lupin	29–50	320–339	355–405	[57,238]
Field pea	3.6–59	130–322	146–347	[31,89,90,114,233,238,240,241]
Soybean meal	12–139	141–231	159–303	[233,240,242,243]

**Table 24 animals-14-00619-t024:** Nutritional value (g/kg, dry matter basis unless otherwise specified) of whole and dehulled faba beans, lupins and field peas.

Nutrient/ANF	Faba Beans ^1,^*		Faba Beans ^2^		ASL ^3^		ASL ^2^		White Lupins ^4,#^		Field Peas ^5^		Field Peas ^2^	
	W	D	W	D	W	D	W	D	W	D	W	D	W	D
Dry matter	888	885	883	879	911	906	9301	896	-	-	906	925	869	874
Crude protein	289	321	275	322	250	311	309	416	430	518	208	234	232	256
Crude fat	-	-	-	-	54	65	-	-	107	119	12	10	-	-
Crude fibre	95	26	-	-	-	-	-	-	140	44	-	-	-	-
NDF	168	85	-	-	290	130	-	-	-	-	142	129	-	-
ADF	126	27	-	-	219	82	-	-	-	-	64	22	-	-
Starch	-	-	408	464	-	-	-	-	-	-	403	465	464	481
NSP (total)	-	-	205	98	-	-	495	259	-	-	-	-	160	92
Soluble NSP	-	-	20	14	-	-	32	19	-	-	-	-	19	15
Insoluble NSP	-	-	185	84	-	-	463	240	-	-	-	-	141	77
Ash	46	45	-	-	38	39	-	-	40	39	27	28	-	-
Calcium	-	-	-	-	3.6	2.4	-	-	-	-	0.7	0.3	-	-
Phosphorous	-	-	-	-	6.1	7.7	-	-	-	-	3.0	3.2	-	-
Tannins	4.4	2.6	-	-	-	-	-	-	-	-	-	-	-	-
TI activity (TIU/mg)	4.2	6.5	-	-	-	-	-	-	-	-	-	-	-	-
Indispensable amino acids														
Arginine	8.7	8.9	24.5	29.5	24.2	28.7	25.4	36.5	9.4	10.1	18.7	20.1	17.4	19.0
Histidine	2.8	2.9	6.6	7.7	6.8	8.5	7.9	10.3	4.8	3.6	5.1	5.5	5.6	6.1
Lysine	6.3	6.1	15.3	17.4	12.2	14.3	14.5	18.6	5.0	5.7	15.9	17.1	16.1	17.0
Phenylalanine	4.1	4.2	10.3	12.2	9.9	12.1	10.9	15.0	2.1	1.1	10.8	11.2	10.5	11.2
Leucine	7.4	7.7	17.7	21.0	16.6	21.4	18.8	25.9	5.8	4.0	15.2	16.5	15.0	16.0
Isoleucine	4.3	4.4	9.4	11.1	9.9	12.3	10.4	14.4	2.3	1.6	7.4	8.3	8.4	8.9
Valine	4.8	5.1	24.5	29.5	10.6	11.8	25.4	36.5	3.2	3.4	8.7	9.7	17.4	19.0
Methionine	0.7	0.8	1.9	2.1	1.6	2.3	1.9	2.6	4.8	5.5	1.9	2.0	2.0	2.0
Threonine	3.4	3.0	8.4	9.7	9.2	10.5	10.2	14.1	-	-	8.2	8.7	7.8	8.1
Tryptophan	0.7	0.8	-	-	2.4	3.1	-	-	4.7	5.1	-	-	-	-

^1^ cv. Alfred [24]; ^2^ cv. PGG Tic [31]; ^3^ cv. Boregine [250]; ^4^ cv. Amiga [251]; ^5^ cv. Trapper [102]. Abbreviation: ANF—antinutritional factor; W—whole; D—dehulled; ASL—Australian sweet lupin; NDF—neutral detergent fibre; ADF—acid detergent fibre; NSP—non-starch polysaccharides; TI—trypsin inhibitor; TIU—trypsin inhibitor units; na—not available. * Air dry basis; ^#^ Amino acid concentrations are based on g per 16 g nitrogen.

**Table 25 animals-14-00619-t025:** The concentration of nutrients and selected antinutritional factors (g/kg, dry matter basis unless otherwise specified) of raw and extruded grain legumes.

Nutrient/ANF	Chickpeas ^1,^*		Sweet Lupins ^2^		White Lupins ^3^		Field Peas ^4^		Faba Beans ^5,^*	
	Raw	Ext.	Raw	Ext.	Raw	Ext.	Raw	Ext.	Raw	Ext.
Dry matter	866	932	-	-	-	-	-	-	864	841–861
Crude protein	200	208	279	311	369	377	230	229	239	232–236
Crude fat	135	68	59	60	130	136	25	26	-	-
Crude fibre	64	65	173	161	-	-	-	-	-	-
Ash	35	36	33	34	37	36	31	31	37	37–38
ADF	-	-	249	229	-	-	-	-	117	110–120
NDF	-	-	271	245	-	-	-	-	156	127–139
NSPs	-	-	-	-	-	-	-	-	-	-
Total	-	-	-	-	371	379	200	194	-	-
Soluble	-	-	-	-	41	74	23	28	-	-
Insoluble	-	-	-	-	330	305	177	166	-	-
Starch	-	-	-	-	nd	nd	465	461	367	354–368
Resistant starch	-	-	-	-	-	-	-	-	192	10–15
AME (MJ/kg DM)	10.8	11.5	8.6	7.5	9.9	7.8	11.7	11.1	7.9^±^	9.0–10.9 ^±^
Amino acids										
Arginine	13		27	32	36	32	-	-	24	22–23
Histidine	2.0	1.9	7.6	8.3	9.1	8.6	-	-	6.8	6.3–6.8
Lysine	15	15	12	13	19	17	-	-	18	17–18
Phenylalanine	13	14	10	11	18	17	-	-	10	9.2–10
Leucine	19	19	18	19	30	32	-	-	19	17–19
Isoleucine	9.6	9.4	10	12	16	15	-	-	10	9.6–10
Valine	12	11	11	11	16	17	-	-	11	10–11
Methionine	2.8	2.5	1.5	1.6	3.8	3.7	-	-	0.5	0.4–0.7
Threonine	9.3	94	9.2	10	15	15	-	-	9.0	8.5–9.3
Tryptophan	-	-	2.3	2.5	-	-	-	-	-	-
Antinutritional factors										
TIA (mg/g)	-	-	-	-	1.1	0.3	0.23	0.19	1.04	0.76–0.80
Phytic acid (g/kg)	-	-	-	-	-	-	-	-	0.47 ^#^	0.42–0.43 ^#^
Tannins (g/kg)	-	-	-	-	-	-	-	-	-	-

^1^ Variety Kabuli [130]; ^2^ cv. Wonga [14]; ^3^ cv. Kiev mutant [74]; ^4^ [89]; ^5^ cv. Amulet [107]. Abbreviations: ANF—antinutritional factors; Ext.—extruded; ADF—acid detergent f ibre; NDF—neutral detergent fibre; NSPs—non-starch polysaccharides; AME- apparent metabolisable energy; TIA—trypsin inhibitor activity. * As-fed basis; ^±^ nitrogen-corrected AME; ^#^ phytate phosphorous.

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
