# Peer review of "Feeding Value of Lupins, Field Peas, Faba Beans and Chickpeas for Poultry: An Overview"

_animals, 2024, doi:10.3390/ani14040619_

Round 1
Reviewer 1 Report
Comments and Suggestions for Authors
Editor decision
This review briefly describes the morphology and nutritional value of lupins, field beans, faba beans and chickpeas, gives examples of protein and non-protein antinutrients and non-starch polysaccharides in antinutrients, and points out that physical, chemical and biological processing can damage antinutrients in legumes. Based on the current literature in the field, the reasons why lupins, field beans, faba beans and chickpeas can replace soybean meal as protein sources in poultry diets were compared and analyzed, and the negative effects of anti-nutritional factors could be reduced through effective processing methods. The content of this article is solid and the literature is fully cited, but there are still problems that need to be further revised and improved.
1. Line 116: Supplementary description of the morphology of Lupins angustifolius.
2. Line 149: What is the response to low tryptophan content in sweet lupine? Or what are the dilemmas of raising tryptophan levels? The literature needs to be supplemented.
3. Line 158: Gungurru has a higher AME value, can we pay more attention to it and discuss it in more detail?
4. Line 227: Can the deficiency of methionine, cysteine and tryptophan in white lupine be improved by modern breeding techniques? If not, what is the problem? Please make a brief literature supplement.
5. Line 273: These reports indicate that the addition of white lupine has a negative effect on broiler production due to alkaloid content >0.1g/kg. Please supplement the literature.
6. Line 453: For the starch and fiber content of Desi and Kabuli varieties, please supplement the numerical range of the specific literature.
7. Line 462: Whether the deficiency of methionine and cysteine in chickpeas can be resolved, please supplement the literature.
8. Line 611: Brief examples of non-toxic and toxic lectins.
9. The distribution and content characteristics of protease inhibitors, lectins, tannins and phytic acids in lupins, field beans, faba beans and chickpeas were added.
10. In the fermentation section of processing, the effects of fermentation on the nutritional value and anti-nutritional factors in lupins, field beans, faba beans and chickpeas were added.
Comments on the Quality of English Language
Fluent language without obvious errors.
Author Response
Animals 2873290 Author responses
The comments are appreciated. All are useful and favourably considered in the revision. The changes are highlighted in YELLOW.
Reviewer 1
This review briefly describes the morphology and nutritional value of lupins, field beans, faba beans and chickpeas, gives examples of protein and non-protein antinutrients and non-starch polysaccharides in antinutrients, and points out that physical, chemical and biological processing can damage antinutrients in legumes. Based on the current literature in the field, the reasons why lupins, field beans, faba beans and chickpeas can replace soybean meal as protein sources in poultry diets were compared and analyzed, and the negative effects of anti-nutritional factors could be reduced through effective processing methods. The content of this article is solid and the literature is fully cited, but there are still problems that need to be further revised and improved.
- Line 116: Supplementary description of the morphology of Lupins angustifolius.
Response: Supplementary description is added as requested (Lines 119-121).
- Line 149: What is the response to low tryptophan content in sweet lupine? Or what are the dilemmas of raising tryptophan levels? The literature needs to be supplemented.
Response: In practice, there is no problem because of the use of synthetic tryptophan to balance the requirement. But there are no published data investigating tryptophan supplementation in legume-based diets for poultry. However, some statements are added (Lines 154-156). Apart from this, a minor revision has been made in Table 3 (for reference [26]) as the values for tryptophan and valine were interchanged.
- Line 158: Gungurru has a higher AME value, can we pay more attention to it and discuss it in more detail?
Response: Based on Table 4, this could be true as the upper range of AME value of Gungurru is having the highest value of 11.64 MJ/kg. However, a lower value of 6.53 MJ/kg is also mentioned as a lower range for Gungurru. In addition, there can be differences in the assay methodologies between the studies. Therefore, based on the current literature, I think, there is not much evidence to say that Gungurru has a higher AME when compared to others. Apart from this, another minor change has been done (based on Table 4) for the lower range of AME as highlighted in line 157.
- Line 227: Can the deficiency of methionine, cysteine and tryptophan in white lupine be improved by modern breeding techniques? If not, what is the problem? Please make a brief literature supplement.
Response: It may be possible, but there is no published data on cysteine or tryptophan. See added statements (Lines 235-238).
- Line 273: These reports indicate that the addition of white lupine has a negative effect on broiler production due to alkaloid content >0.1g/kg. Please supplement the literature.
Response: See revisions in Lines 277-281.
- Line 453: For the starch and fiber content of Desi and Kabuli varieties, please supplement the numerical range of the specific literature.
Response: Thank you for the comment. Numerical range based on specific literature is now supplemented separately as requested (Lines 454-457).
- Line 462: Whether the deficiency of methionine and cysteine in chickpeas can be resolved, please supplement the literature.
Response: Revised as suggested –see L467-9.
- Line 611: Brief examples of non-toxic and toxic lectins.
Response: Examples are added as requested (Lines 628-630).
- The distribution and content characteristics of protease inhibitors, lectins, tannins and phytic acids in lupins, field beans, faba beans and chickpeas were added.
Response: Sorry - the comment is not clear. There was no PDF attached to the referee’s report. Also, as indicted in this review, exhaustive literature is available and a detailed discussion is beyond the scope of the paper.
- In the fermentation section of processing, the effects of fermentation on the nutritional value and anti-nutritional factors in lupins, field beans, faba beans and chickpeas were added.
Response: Again the comment is not clear. However, some additional information on the effect of fermentation on nutritional value and anti-nutritional factors was added for specific grain legumes (Lines 920-923).

Reviewer 2 Report
Comments and Suggestions for Authors The authors reviewed the feeding values of four grain legumes, which are valuable for the feed and poultry industry. However,there are some issues that should be addressed before acceptance. The title does not match the topic in the MS which mainly focuses on the nutritional values and antinutritional factors. Please revise the title. Or reorganize the MS just focus on some key parts. L129-130 Are nutrition values affected by the culture season or processing method? Please provide some references. The same issues also apply to the other beans in the remaining MS. Table 2,6,10,14,18 . What about the average or mean value? Please refer to the statistical method. 2.1.1.3 Feed trail A table include the broliler breed (AA, ROSS, cobb, or other breeds), trail lasted time, inclusion level of lupin and the approaite levels suggested in the reference can be listed here. The same issues were presented in the other beas of the MS, for example. 2.1.2.3, L223-224 Please provide references in this section. L235-236, Table 7, Table 9, Table 11, Table 13, Table 15, Table 19, The species can be listed in the Table, not in the footnote. Table 8. The broiler breed can be included here. L280 What is the age of the turkeys? Table 20, This table format can be adjusted to fill the page. For example, it can be displayed in either four or six columns. The section discusses "3. Antinutritional factors in grain legumes." and "4. Improving the feeding value of grain legumes". The definition and processing of antinutritional factors were discussed extensively, which deviated from the theme of the MS. The effects of these antinutritional factors on poultry feeding should be discussed here, in line with your title and the main topic. As for the concluding remarks, this part should be more concise. The appropriate inclusion levels may vary for different poultry species. The suggested inclusion level of "200-300 g/kg" may not be rigorous enough. Furthermore, the future applications of these grain legumes can be outlooked.Author Response
The title does not match the topic in the MS which mainly focuses on the nutritional values and antinutritional factors. Please revise the title. Or reorganize the MS just focus on some key parts
RESPONSE: Technically, the term ‘feeding value’ encompasses several related parameters – composition, nutrient utilisation, (the effects of) antinutritional factors, strategies to improve the nutrient utilisation etc. Thus the title is retained. We have not considered the second option - reorganising the approach of the MS will be cumbersome and will require complete rewriting.
L129-130 Are nutrition values affected by the culture season or processing method? Please provide some references. The same issues also apply to the other beans in the remaining MS.
RESPONSES: Relevant references are shown in the manuscript. See L135.
Table 2,6,10,14,18 . What about the average or mean value? Please refer to the statistical method.
RESPONSE: In contrast to those reporting proximate and carbohydrate composition, only limited reports are available for the parameters (AA, AA digestibility etc.) presented in these Tables. Hence we have chosen not to do averages, but rather provide the individual study data.
2.1.1.3 Feed trail A table include the broliler breed (AA, ROSS, cobb, or other breeds), trail lasted time, inclusion level of lupin and the approaite levels suggested in the reference can be listed here. The same issues were presented in the other beas of the MS, for example. 2.1.2.3,
RESPONSE: We believe that additional tables will not add further value. For example, data from 8 studies were used in 2.1.1.3. Four of them do not identify the broiler strain. The growth phases (starter/ grower/ finisher/ overall) as well as recommended inclusion levels are already identified in the submission.
L223-224 Please provide references in this section.
RESPONSE: Provided as suggested. See L 223
L235-236, Table 7, Table 9, Table 11, Table 13, Table 15, Table 19, The species can be listed in the Table, not in the footnote.
RESPONSE: The suggestion is appreciated, but we feel that use of foot notes is equally effective.
Table 8. The broiler breed can be included here.
RESPONSE: No changes made. Strain is identified in some assays.
L280 What is the age of the turkeys?
RESPONSE: Good point - 0-21d; Revised now. See L286-287.
Table 20, This table format can be adjusted to fill the page. For example, it can be displayed in either four or six columns.
RESPONSE: not clear. This table (for chickpeas) has only one reference/digestibility value.
The section discusses "3. Antinutritional factors in grain legumes." and "4. Improving the feeding value of grain legumes". The definition and processing of antinutritional factors were discussed extensively, which deviated from the theme of the MS. The effects of these antinutritional factors on poultry feeding should be discussed here, in line with your title and the main topic.
RESPONSE: As indicated for comment# 1, these two aspects are very closely entwined with the nutrient utilisation and, hence the nutritional/feeding value of these legumes. These are already discussed under these sub-sections in this overview.
As for the concluding remarks, this part should be more concise. The appropriate inclusion levels may vary for different poultry species. The suggested inclusion level of "200-300 g/kg" may not be rigorous enough. Furthermore, the future applications of these grain legumes can be outlooked.
RESPONSE: Agree with comment that the inclusion levels will differ for different species. But there are not enough published data currently available to be more specific.

Reviewer 3 Report
Comments and Suggestions for Authors
Dear authors,
Your work represents an interesting and really extensive literature review investigating the potential use of legume seeds as an alternative protein source while considering any anti-nutritional factors.
Reading your manuscript, some details need revisions.
1. Please, replace "alleviate" with "mitigate" (Line 18).
2. Please, replace "is due largely to differences" with "is largely caused by" (Line 129).
3. When you are referring to aminoacid in the text and in Tables, please consider to replace "indispensable" with "essential" and "dispensable" with "non-essential". These are the most commonly used forms.
4. In Tables containing literature references, please replace column title "reference" with "references".
5. In the text you mention numerous cultivars referring to the plants you extensively describe (for example Table 4 or Line 177). Please, pay attention you are referring to such specifically selected plants. That is, check whether it is correct in all cases to speak about cultivars or whether it would not be more appropriate to speak about varieties.
I mean that "cultivar" is purposely created by humans to enhance chosen traits through artificial selection, "variety" is a version of the plant species that occurs naturally through natural selection and need the italics when you are reffering to it.
6. Please, erase the literature references reported in the conclusions (Line 1081).
Comments on the Quality of English LanguageMinor editing are required.
Author Response
The comments are appreciated. All are useful and favourably considered in the revision. The changes are highlighted in YELLOW.
Reviewer 2
Your work represents an interesting and really extensive literature review investigating the potential use of legume seeds as an alternative protein source while considering any anti-nutritional factors.
Reading your manuscript, some details need revisions.
- Please, replace "alleviate" with "mitigate" (Line 18).
Response: Replaced as requested.
- Please, replace "is due largely to differences" with "is largely caused by" (Line 129).
Response: Replaced as requested.
- When you are referring to amino acid in the text and in Tables, please consider to replace "indispensable" with "essential" and "dispensable" with "non-essential". These are the most commonly used forms.
Response: Replaced as requested.
- In Tables containing literature references, please replace column title "reference" with "references".
Response: Replaced as requested.
- In the text you mention numerous cultivars referring to the plants you extensively describe (for example Table 4 or Line 177). Please, pay attention you are referring to such specifically selected plants. That is, check whether it is correct in all cases to speak about cultivarsor whether it would not be more appropriate to speak about varieties.
I mean that "cultivar" is purposely created by humans to enhance chosen traits through artificial selection, "variety" is a version of the plant species that occurs naturally through natural selection and need the italics when you are referring to it.
Response: Good point and thanks. Revised as appropriate.
- Please, erase the literature references reported in the conclusions (Line 1081).
Response: Deleted as requested.

Round 2
Reviewer 2 Report
Comments and Suggestions for Authors
hank you for your response. I agree with your viewpoint and explanation. Overall, the manuscript holds significant value for the feed industry.